

# Vegetation vulnerability to drought on southeastern Europe

Patrícia Páscoa[1], Célia M. Gouveia[1,2], Ana C. Russo[1], Roxana Bojariu[3], Sergio M. Vicente-Serrano[4] and Ricardo M. Trigo[1]

[1]Instituto Dom Luís, Faculdade de Ciências da Universidade de Lisboa, Campo Grande, Edifício C8, Piso 3, 1749-016 Lisboa, Portugal
[2]Instituto Português do Mar e da Atmosfera, Lisboa, Portugal
[3]National Meteorological Administration, Bucharest, Romania
[4]Instituto Pirenaico de Ecología, CSIC, Zaragoza, Spain

*Correspondence to:* Patrícia Páscoa (ppramos@fc.ul.pt)

**Abstract.** In this work, the impacts of drought events on vegetation activity in the period 1998-2014 were analysed, on an area located in southeastern Europe, using the Standardized Precipitation Evaporation Index (SPEI) at several time scales, and the Normalized Difference Vegetation Index (NDVI), as obtained by SPOT/VEGETATION. A correlation analysis between monthly NDVI and SPEI was performed, as well as the assessment of the simultaneous occurrence of drought events and low vegetation activity. The analysis was performed from April to October, corresponding to the months showing the highest vegetation activity. The impacts of drought were assessed in the entire period, but also on the drought event in 2000/2001, which was the driest in this area since 1961. The results show the area with positive correlation between the two indices reaches 77% in July on agricultural land, and its impacts are seen from May to October, pointing to a high vulnerability to drought. Forests seems to be less sensitive to drought, although being more influenced by drought mostly on July and August. However, negative correlations were also observed in June at high altitudes pointing to a positive effect of dryness on the vegetation activity. This feature changed in July, possibly as a result of the increasing temperature coupled to the decrease in precipitation that leads to a water stress situation. The drought event in 2000 provoked a generalized decrease in vegetation activity, and 71% of the area showed a decrease on at least four months between April and October. Almost 60% (20%) of the studied area presents a concordance of anomalous low vegetation activity and dry conditions of at least 50% (80%) on the month of August, whereas the Carpathian Mountains generally present a lower value of concordance than the remaining area.

## 1 Introduction

Natural disasters occur frequently and play a significant role on society and ecosystems (Kelman et al., 2016), having strong economic, environmental and social impacts. Drought is one of the most important hydroclimatic hazards affecting society and the environment (Below et al., 2007; Carrão et al., 2016; Wilhite & Pulwarty, 2017). Droughts are difficult to quantify, as it can be tricky to determine the onset, end, severity and spatial extent of the drought event (Wilhite, 2000). Moreover, given the number of socioeconomic sectors and environmental systems affected, the establishment of a simple and universal definition of drought events is impossible (Lloyd-Hughes, 2014). Drought strongly affects human activities such as agriculture





(Li et al., 2009; Lesk et al., 2016), power production (Stagge et al., 2015),  water supply (de Wit & Stankiewicz, 2006; Rajagopalan et al., 2009), as well as natural ecosystems (Allen & Breshears, 1998; Zeng et al., 2005, Fauset et al., 2012;), causing episodes of tree mortality (Allen et al., 2010), reducing the primary and secondary tree production (Martínez-Alonso

et al., 2007, Bennet et al., 2015) and the productivity of pasture lands (Smit et al., 2008). The identification of drought impacts on natural systems is complex given the varied resistance and resilience of vegetation types to water deficits (Chaves et al., 2003; McDowell et al., 2008; Gazol et al., 2018) and also the complex interactions between drought severity and drought time scales to explain drought effects on vegetation (Vicente-Serrano et al., 2013 and 2014; Gouveia et al., 2017).

During the last decades the assessment of drought impacts on vegetation has been done through different approaches: i) the

analysis of the hydraulic properties of vegetation, on a species or biomes level (Anderegg et al., 2012; Choat et al., 2012), ii) the use of tree-ring data to assess the impact of droughts on tree growth (Rigling et al., 2002; Bigler et al., 2006;), iii) the analysis of crop yields (Hlavinka et al., 2009; Páscoa et al., 2017), and iv) phenological information (Bernal et al., 2011; Ivits et al., 2014). Furthermore, remote sensing has emerged as a powerful tool to monitor vegetation as it has the ability to cover large areas with repeated measurements over time and high spatial resolution (Kerr and Ostrovsky, 2003). The impacts of

drought and climate variability on crop yields and vegetation activity have been successfully assessed using different vegetation indices such as the Normalized Differences Vegetation Index (NDVI) on local, regional and global scales (Vicente-Serrano, 2007; Vicente-Serrano et al., 2015; Mkhabela, et al., 2011; Xu et al., 2012; Gouveia et al., 2017).

On the other hand, multiscalar drought indices such as the Standardized Precipitation Index (SPI, McKee et al., 1993) and the Standardized Evapotranspiration Index (SPEI, Vicente-Serrano et al., 2010), which are computed using meteorological

variables, allow to identify drought impacts on a variety of vegetation types and biomes. This flexibility results from the fact that these indices are able to adapt to the different times of response of vegetation to water deficit conditions (Pasho et al., 2012; Levesque et al., 2013; Vicente-Serrano et al., 2013; Gouveia et al., 2017) and solve the common problems existing to relate ecosystem variables with drought severity.

The Mediterranean basin is one of the main world regions affected by drought, although the ability of the Mediterranean

vegetation to adapt to drought is well known (Lindner et al., 2010). However, changes in landscape imposed by many centuries of human management are affecting the capacity of vegetation to cope with drought (Puigdefábregas & Mendizábal, 1998; Alados et al., 2011). In addition, the effect of drought on vegetation exhibits distinct patterns in different regions of the Mediterranean basin, both in terms of time scales and types of vegetation affected (Gouveia et al., 2017, Bento et al., 2018), being necessary high spatial resolution studies that allow to identify the existing diversity of the drought impacts on vegetation.

Southeastern Europe (SEE) is a highly diverse region located between the Mediterranean Sea and the Black Sea, with highly varied bioclimatic characteristics and vegetation types (Horvat et al. 1974; Donita et al. 1992, Bohn et al., 2004), having areas with a very intense land use during centuries and altered natural vegetation (e.g. Badea & Biris 2012; Knorn et al. 2012). However, the land use modifications have some distinct characteristics when compared with the Western Europe (Kuemmerle et al. 2016, Feurdean et al. 2017). For instance, grasslands have persisted longer here, as agricultural intensification was

delayed in comparison with Western Europe (Jepsen et al. 2015; Hajnalova & Dreslerova 2010). In the period 1947–1989, the





transfer of land ownership to the state led to agricultural intensification in many areas (Sârbu et al. 2004), and after 1989, this trend has reversed, resulting in farmland abandonment (Kuemmerle et al. 2009; Stoate et al. 2009). A large percentage of the forests in the region is monoculture plantation or semi-natural forest (Csagoly, 2007), which have a lower resilience to climate variability than natural mature forests (Noss, 2001). Nevertheless, there are also well preserved natural ecosystems: i.e. the

largest pristine forests in central and eastern Europe are located in the Carpathian Mountains, which are very rich in biodiversity (Csagoly, 2007). On the other hand, grasslands and agricultural mosaics in SEE are also rich in biodiversity (Loos et al., 2015), and some rare or endangered species are found in grasslands in this area, such as several species of butterflies (Cremene et al., 2005). Although the aridity classification in SEE ranges from humid to dry-subhumid (Cheval et al., 2017), the annual water balance is negative on territories dominated by agriculture, and this value decreases during the growing season

(Paltineanu et al., 2007).

Climate trends in the past decades have identified relevant changes such as increasing temperature (Dumitrescu et al. 2014, Marin et. al. 2014) and like increases in the Atmospheric Evaporative Demand (AED) (Croitoru et al., 2013), which may have impacts on the vegetation primary and secondary growth (Pravalie et al., 2014). Moreover, climate change models predict increased climate impacts in SEE, which are associated to a precipitation decrease in the warm season (Bojariu et al. 2015)

but also to the increase of the AED (Cheval et al., 2017). This scenario also implies the increase in the climate aridity and more severe and long drought events, with relevant impacts on the ecosystems of the region. This makes it especially relevant to evaluate in more depth the different impacts of the current biomes to drought events, highlighting the need to understand the relation between vegetation activity and climate/environmental characteristics in the region.

In this study we analyse the impacts of drought severity on vegetation activity obtained from satellite information in the SEE

using NDVI and SPEI at different time scales. The objectives were: i) to map the vegetation response to drought conditions, ii) to identify the vegetation types that show the strongest response to drought severity and iii) to study the impact of a severe drought episode on the vegetation activity.

## 2 Data and methods

### 2.1 Land cover and elevation data

Land cover classification was retrieved from the Globcover 2009 project (http://due.esrin.esa.int/page_globcover.php). It has a spatial resolution of 300m and contains 22 land cover categories. The Globcover 2009 map was resampled to the resolution and coordinates of NDVI by means of a majority rule, followed by a nearest neighbour interpolation. Water bodies were eliminated and land cover classes covering a small area were also discarded, in order to analyse the land covers that are more representative of the study domain. Overall the discarded land cover classes occupy less than 1% of the total land area.

Elevation data was obtained from the GLOBE project (GLOBE Task Team, 1999) at a spatial resolution of 1km, and it was resampled to the coordinates of NDVI using a bilinear interpolation.



The percentage area occupied by the remaining land cover classes is shown on Table 1. More than two thirds of the considered area (70.6%) is occupied by agricultural land, classified as rainfed cropland (34.8%), or a mixture of cropland and natural vegetation (26.2 and 9.6%). Three classes of forests are present, occupying 23.7% of the area, and a mixture of grassland and forest or shrubland occupies 3.8%. Table 1 also presents the labels used afterwards in Figs. 1, 3, 4, 5, and 10, to represent each land cover class. Figure 1 maps these land cover classes (top panel), being evident the distinct shape of the Carpathian Mountains and therefore the dependence of land cover on the topography. Agricultural areas are the dominant land cover at lower altitudes (Fig.1, bottom left panel), whereas forest areas are more common at higher altitudes and become the dominant land cover type at altitudes higher than 500m (Fig.1, bottom left panel). The characteristics of the forest also changes with altitude, and so the broad-leaved forest is more common at lower altitudes, whereas needle-leaved forest, although only accounting for 3.3% of the total area, covers the largest area at altitudes higher than 1000m. It should be noted in any case that, despite the high altitudes reached in this mountainous region, more than half of the study area (56.5%) sits below 250m (Fig.1, bottom left panel).

## 2.2 NDVI data

There are currently several NDVI datasets which differ in the spatial and temporal resolution and the time period covered. The GIMMS dataset provides NDVI time series with an 8 km spatial resolution since 1981, and it is the longest NDVI time series covering the study area. More recent datasets have a higher spatial resolution, although the time period is inherently shorter. Examples are NDVI obtained by the SPOT VEGETATION instrument, which has a spatial resolution of 1 km, and by MODIS instrument, which has a spatial resolution of 250m. Considering the large extent of the study area and the previous utilization of the dataset for vegetation dynamic and drought assessment for similar studies at regional scale (Gouveia et al., 2009, Trigo et al., 2010, Barriopedro et al, 2012), the spatial resolution of SPOT VEGETATION was considered as the most appropriate and so this dataset was used (https://www.vito-eodata.be/PDF/portal/Application.html). The time series covers the period from April 1998 to May 2014 and includes one value per 10 days, retrieved using the Maximum Value Composite (MVC) technique (Holben, 1986). NDVI from this dataset was computed using atmospherically corrected and geometrically calibrated data. A monthly NDVI time series was obtained by selecting the highest value of each month. Pixels identified as snow cover by the status map were excluded and values below 0.1 were discarded, since they do not represent vegetation.

The median monthly values of NDVI for each land cover type are shown on Fig. 1 (bottom right panel). In the winter months, the vegetation activity is lower for all land cover types, and it increases significantly in April and May. June is the month showing the highest NDVI value for almost all land cover types, with the exception of needle-leaved forest, which shows the highest NDVI value on August. In the month of November, the NDVI values decreased sharply on all land cover types. Considering that the highest signal of drought variability is expected to be found on the months showing the highest vegetation activity (Vicente-Serrano et al., 2015), the analysis will be performed between the months of April and October.





## 2.3 Drought index

Drought conditions and variability were assessed using the Standardized Precipitation Evaporation Index (SPEI), which has been shown to be particularly suitable to characterize vegetation and hydrological drought conditions (Vicente-Serrano et al., 2010, 2012, 2013). The necessary precipitation and AED to calculate the SPEI was retrieved from the Climate Research Unit (CRU TS3.23) database, which has a 0.5º spatial resolution and covers the period 1901-2014 (Harris et al., 2014). SPEI was computed for the 1-, 3-, 6-, 9-, and 12-month time scales, since it is expected that different vegetation types will respond to different SPEI time scales as shown before for the entire Mediterranean basin (Gouveia et al. 2017). A threshold of -0.84 was established to identify the drought events. This value corresponds to the most severe drought event expected in a period of 5 years (Agnew, 2000).

## 2.4 Analysis

### 2.4.1 Correlation analysis

This analysis will follow a similar approach to Gouveia et al. (2017), which have examined the drought impacts on vegetation on the entire Mediterranean basin and including the area studied here, using SPEI and NDVI from GIMMS dataset at 8-km spatial resolution. Nevertheless, the 1-km spatial resolution of NDVI obtained from SPOT-VGT dataset used in this work, together with the use of land cover information, allows for a more detailed understanding of vegetation vulnerability to drought and its spatial variability than the former. Thus, a correlation analysis between monthly NDVI and SPEI was performed, using the two-tailed Pearson correlation coefficient and a level of significance of 0.1. Both time series were detrended prior to the computation of the correlation in order to avoid potentially misleading interpretations of results amplified by possible trends in the time series as a consequence of changes in drought severity (Sousa et al., 2011; Dai, 2013), improvements in agricultural technologies (Calderini & Slafer, 1998; Hafner, 2003) and/or CO2 fertilization influences (Piao et al., 2006; Donohue et al., 2013). SPEI time series was rescaled to match the NDVI resolution, using a bilinear interpolation.

In order to assess the impact of drought on each land cover type, the percentage of area showing significant positive and negative correlations was assessed for all SPEI time scales. The mean value and the standard deviation of the significant correlations on each land cover type was also computed, for each month and time scale. As shown on Fig. 1, the occurrence of a land cover type is dependent on the elevation, which could also influence the results obtained. Therefore, the percentage area showing significant correlations along the altitude range was assessed, for each land cover type and SPEI time series, on the months of June and July, with the aim of confirming that the results were related to the different land cover types, and not solely related to the elevation range.

### 2.4.2 Occurrence of vegetation stress and drought

A correlation analysis between NDVI and SPEI reflects the interannual variability of climate conditions, which includes dry and wet events. In order to assess the effects of drought conditions separately, the simultaneous occurrence of low NDVI





anomalies and drought conditions, identified when SPEI values are lower than -0.84, was computed as a percentage relative to the total number of NDVI anomalies lower than -0.025 (this threshold is discussed in the next section). This analysis was performed for the entire period, for each month and time scale.

### 2.4.3 Drought episode of 2000/2001

Several droughts have been diagnosed in recent decades for the study area. Nevertheless, the drought episode of 2000/2001 was identified as the most severe in the period 1960-2013 (Spinoni et al., 2013, Ionita et al. 2016,). Using 6-months SPI and SPEI, Spinoni et al. (2013) identified the start of this drought event in June 2000, and the end in February 2001. The methodology proposed by Gouveia et al. (2009) was used to assess the impacts of this drought episode on the vegetation activity. NDVI monthly anomalies were computed, after subtracting the NDVI value obtained for each month from the

corresponding median. This methodology was applied on the drought months of 2000 only, since the drought months of 2001 coincide with the months of lowest vegetation activity. As described in several previous applications for different areas of the globe, persistent negative NDVI anomalies are symptomatic of vegetation stress (e.g. Gouveia et al., 2009; Trigo et al., 2010; Barriopedro et al., 2012). Sensitivity tests were performed with the aim to define the adequate threshold to be used to assess the stress conditions using NDVI anomalies.  Considering the land cover types present in the study area (Fig.1), the vegetation

was identified as under stress conditions if NDVI anomalies were lower than -0.025, since this threshold was used successfully on an area with similar vegetation types (Gouveia et al., 2009, 2012). Thus we computed and mapped the number of months showing NDVI anomalies lower than -0.025, during the peaks of vegetation activity. The number of months with SPEI values below -0.84 was also computed and mapped for the same time period, with the purpose of assessing the spatial extent of the drought episode at various time scales and the overlapping of vegetation under stress conditions and the occurrence of drought

conditions. The percentage area of each land cover class showing vegetation stress was also computed, as well as the corresponding duration, allowing to identify the response of different land cover types to drought severity.

## 3 Results

### 3.1 Correlation between NDVI and SPEI

Figure 2 shows the correlations between NDVI and SPEI for the time scales of 1, 3, and 6 months, from April to October. For

the sake of simplicity and taking in account that the results obtained with the time scales of 9 and 12 months are very similar with the time scale of 6 months, we opted to exclude those two time scales from the results analysis. The percentage of surface area characterised by significant positive and negative correlations on each land cover type is shown on Fig. 3. Negative correlations are more frequently found on the months of May and June, and are mostly located in the Carpathian Mountains, although there are positive correlations in these months as well (Fig. 2). The land cover types showing a larger area of negative

correlations in this period are both the needle-leaved and the broad-leaved forests, with areas reaching around 70% and 35%,



respectively, in June, for the time scale of 1 month. The land cover type of mixture of grassland and forest or shrubland also exhibits a considerable percentage area with negative correlations (around 25%) on the same month and time scale.

During summer months the significant correlations are mostly positive and dominate a larger area at SPEI time scales of 6 months. At shorter time scales, the study area shows no significant correlation on the area south of the Carpathian Mountains in June. Nevertheless, on July and August, this area is positively correlated with SPEI, whereas the Carpathians and the northern areas reveal less significant correlations (Fig. 2).

The areas showing positive correlations from April to June are mostly agricultural land (Fig.3), and the surface area showing significant positive correlations increases with the SPEI time scale, exceeding 37% of the total cropland area in June. From July onwards, the percentage area with positive correlations for SPEI time scales of 6 is very high (Fig. 3) reaching 76% on the case of cropland (in August) and 65% on the case of grassland (in September). For broad-leaved forests this value reaches 42% (in August, for 6 months), and similarly to the agricultural and grassland areas, the areas with positive correlations are particularly widespread during the months of July, August and September. On the other hand, in the case of needle-leaved forests, the surface area with significant positive correlations only reaches 17% in July (for 6 months), and even less for the remaining months (Fig.3). The pattern obtained for the mixture of broad-leaved and needle-leaved forests is very similar to that of needle-leaved forests, although the maximum value obtained in July is only 12%.

The spatial pattern of significant correlations is related to the average altitude, as shown in Fig. 4. This figure depicts the percentage area showing significant negative (positive) correlations in June (July), for the land covers cropland, broad leaved forest, needle leaved forest and grassland along the altitude range, for the time scales of 1 (top panel) and 6 months (bottom panel). The negative (positive) correlations are mostly located at higher (lower) altitudes, and their area decreases with diminishing (increasing) altitude. Needle-leaved forests have the highest area of negative correlations and the lowest area of positive correlations, on all altitude intervals. Cropland presents the lowest area of negative correlations and a large area of positive correlations, although on almost all altitude intervals with positive correlations the land cover type showing the larger area is grassland.

The mean value and the corresponding standard deviation of the significant correlations obtained on each month and time scale for each land cover type are shown on Fig. 5. On agricultural areas, the mean correlation is generally positive, although the standard deviation is lower from July onwards. Despite the area of significant correlations being much smaller for forest land covers, there is a distinct pattern of negative correlations from April to June and of positive correlations from July onwards. Nevertheless, the mean correlations for needle-leaved forest are again negative from August onwards for small time scales. For these land cover types (forests), the standard deviation is lower until July in the needle-leaved and mixed forests.

## 3.2 Occurrence of vegetation stress and drought

The relationship between drought severity and the NDVI was also assessed by counting the number of months characterised by simultaneous occurrence of intense negative NDVI and SPEI anomalies, for each month from April to October and for the SPEI time scales of 1, 3, and 6 months, as the percentage relative to the number of low NDVI anomalies (shown on Fig. 6).



The simultaneous occurrence is dependent on the month and the SPEI time scale, and it also presents spatial variability (Fig.7). August is the month presenting a higher concordance of low NDVI anomalies and low SPEI, and the concordance is high on all time scales (Fig.8). The Carpathian Mountains generally present a lower concordance than the remaining area, and it also presents a lower number of low NDVI anomalies (Fig.6), whereas a higher agreement is found in the central area in May, in the northeastern area in June and in the south in July (Fig.7). There are also some areas with a simultaneous occurrence higher than 50% in May, while for the months of July, September and October, the concordance is much lower at the time scales of 1 and 3 months (Fig.7).

We computed the area showing a concordance of at least 50% (Fig.8, left panel) and 80% (Fig.8, right panel), on each month and time scale. Although April is the month presenting the highest number of NDVI anomalies lower than -0.025 (Fig.6), it is also the month where these low NDVI anomalies coincide with low SPEI on a smaller number of occasions, for all time scales (Fig.8). A 50% concordance is obtained on at least around 40% of the area, from May to October, at time scales of 6 months. For shorter time scales, the area is generally lower. For August, the area reaches 60% on all time scales, but this value is also high in June, July, and September, although at longer time scales. Moreover, the simultaneous occurrence of low NDVI anomalies and low SPEI exceeds 80% on most months: in August this value of concordance is present on 20% of the area, for SPEI at the time scale of 6 months. Such a high concordance is practically absent in April, as well as in September and October at short time scales.

## 3.3 Drought episode of 2000/2001

In this section we assess the impact of the strong drought episode of 2000/2001, the strongest drought episode since the 1950s that have struck this region (Spinoni et al., 2013; Ionita et al., 2016). This goal is achieved looking in depth to the NDVI anomalies between April and October of 2000, corresponding to the period of the highest vegetation activity during the drought. The number of months with NDVI anomalies below -0.025 and SPEI anomalies below -0.84 for the time scales of 1, 3, and 6 months are shown in Fig. 9. The northern region, as well as part of the Carpathian Mountains, were less affected by the drought since these areas present only 3 or less months of low NDVI anomalies (Fig. 9, top left panel). In the remaining areas, corresponding to roughly 70% of the total area, NDVI anomalies were lower than -0.025 for 4 or more months. It is also possible to conclude from Fig. 9 that the SPEI metric identified drought conditions on most of the territory, particularly for the time scale of 6 months and for longer periods of drought (6 and 7 months). The period with monthly dry conditions was shorter in the northern region. Figure 10 presents the percentage area showing NDVI anomalies below -0.025 for different durations (between 0 and 7 months) in each land cover type. The majority of land cover categories showed negative anomalies of the NDVI during several months of the year, but there are noticeable differences. Grassland areas were the most affected, since up to 80% of this area shows negative NDVI anomalies during 4 or more months. The three types of agricultural area show negative NDVI anomalies during 5 or more months on 44% to 51% of the corresponding area, whereas broad-leaved forest showed negative anomalies during 4 or more months on 63% of its area. The needle-leaved forest shows the smallest decrease in vegetation activity, with 59% of its area presenting low NDVI anomalies for only 3 (or less) months.





## 4 Discussion and Conclusions

In this paper, the impacts of droughts on vegetation were analysed, in an area located in southeastern Europe, and the assessment was performed using NDVI and SPEI at several time scales. We identified spatial patterns, as well as drought impacts on the land cover types most common in the study area, which included three classes of cropland, three classes of forest, and one class of grassland. The analysis covered the months of April to October, corresponding to the months of highest vegetation activity as assessed by NDVI. This period is in agreement with the beginning and end dates of the growing season obtained for this area with experimental data (Chmielewski & Rötzer, 2001).

A global analysis of the results obtained for the correlation between NDVI and SPEI allows to identify several important aspects: from April to June, the area with significant correlations increase on all time scales; from June to July the negative correlation disappears almost entirely on all time scales; between July and September, at time scales of 6 months and longer (not shown), the significant correlations occupy most of the territory; and from September onwards the negative correlations reappear, starting at the shorter time scales.

These results are in agreement with the results of other works performed on this area. When studying the drought impacts on several crops on Moldavia, based on SPEI and crop yields, Potopová et al. (2016), found that maize and sugar beet presented a high vulnerability to drought, and the correlation between crop yields and SPEI were also lower in April and highest from July onwards. Levanič et al. (2013), when analysing the relation between Pinus nigra tree-ring widths from southern Romania and monthly precipitation, as well as 3-month SPI, found July to be the month with the highest correlation with precipitation, whereas July, August, and September showed a high correlation with SPI. Moreover, SPEI and NDVI correlations respond to the water balance, and tend to increase when the latter decreases (Vicente-Serrano et al., 2013). In the study area, monthly mean precipitation reaches its maximum in June (Koleva et al., 2008; Cheval et al., 2011; Spinoni et al., 2015), and the maximum temperature occurs in July (Spinoni et al., 2015; Dascălu et al., 2016), which points to increased soil water stress that could explain the increase in the correlations between SPEI and NDVI.

The negative correlations appear mainly at high altitudes, whereas the positive correlations are more frequent at lower altitudes. This pattern is in accordance with the results obtained by Gouveia et al. (2017), although the negative correlations identified by these authors over the considered region are not significant. Moreover, the limiting factors to vegetation activity may change along the altitude range, as a response to the different climate conditions. In the cold and humid mountainous areas, the vegetation activity is more likely to be limited by temperature or radiation (Nemani et al, 2003). Additionally, Babst et al. (2013) showed that, for the latitudinal interval considered in this study, the forest growth is limited by temperature at higher altitudes and by precipitation at lower elevations, and the opposite correlation signals obtained in this work reflect this behaviour. Cannone et al. (2007) found fast growth responses of alpine vegetation to increasing temperatures, since the length of the snow cover season decreased. However, the study of Baumbach et al. (2017) suggests the existence of a nonlinear component in the relation between temperature and vegetation development at high altitudes. In their study, positive temperatures are associated with high NDVI events in the Alps, especially in March-April, but no linear relation between





temperature and NDVI for May and June is apparent in the Carpathians. By analysing the period from May to October, it was possible to identify significant negative correlations at high altitudes in the Carpathians, and also the moment these negative correlations tend to disappear, around July, which coincides with the emergence of significant positive correlations at lower altitudes. Sidor et al. (2015) analysed tree-rings of Norway spruce stands located in the Eastern Carpathians, and found that, at high altitudes, the influence of temperature is positive on June and July, but it is stronger on June, whereas the influence of

precipitation is negative on these months. At lower altitudes these relations are inverted, and the authors suggest that water stress may induce negative responses to temperature. The negative correlations obtained here for the month of June at high altitudes point to a positive effect of dryness on the vegetation activity, and the subsequent signal change in the month of July may be a response to the increasing temperature coupled to the decrease in precipitation, and an indicator of water stress, even though at shorter time scales the correlation at high altitudes is mostly non-significant. At low altitudes, the correlations are

mostly positive, and point to a clear negative effect of aridity on vegetation sensitivity to drought. Moreover, some regions present positive correlations earlier, such as the northeastern region in June, at all the time scales analysed, and the southeastern region, at time scales of 6 months and longer. These areas are characterized by high aridity and low values of water balance (Paltineanu et al., 2007; Croitoru et al., 2013; Mihăilă et al., 2017), making them more vulnerable to drought conditions. Nonetheless, NDVI anomalies indicate that drought events may negatively impact vegetation activity even on areas where the

correlation is negative or non-significant. The relatively small area of significant correlations in the months of April and May point to a small influence of the drought conditions on vegetation activity, whereas the simultaneous occurrence of negative NDVI anomalies and low SPEI reaches 50% in these months. Although the Carpathians region present a negative correlation with SPEI on May, most of this region presents a simultaneous occurrence of drought and negative NDVI anomalies, reaching 50% on some cases. On August, the area of significant correlations increases with the time scale, but it is possible to see high

concordance of drought and low vegetation activity on all time scales, including the intra-Carpathians and the Carpathians regions. Around 60% of the study area presents a simultaneous occurrence of drought and negative NDVI anomalies of at least 50% on this month, pointing to an important influence of drought conditions on vegetation activity which was not as evident when analysing the correlation patterns. Moreover, a concordance of at least 80% is achieved on 10% of the study area at the shortest time scale, but this value reaches 20% for the time scale of 6 months.

Agricultural land appeared to be the most vulnerable to drought, since land cover types associated with cropland present more area with significant positive correlations, and much less area with negative correlations than the remaining land cover types. Moreover, the three land cover types representing agricultural area present a positive mean value of correlation on almost all months and SPEI time scales studied, whereas the land cover types representing forests and grasslands show negative correlations from April to June, and later on September and October. The low values of standard deviation obtained from July

onwards on agricultural land, grassland and broad-leaved forests indicate that these months are the most important on these land cover types. On the other hand, and although needle-leaved and mixed forests also present positive correlations during summer months, the lower area with significant positive correlations and the high values of standard deviation obtained imply that these land cover types are less vulnerable to drought. As stated earlier, the relation between growth of a certain tree species



and climate variables may depend on the altitude (Sidor et al., 2015), and the presence of different land cover types in the region is mostly dependent on elevation. In any case, all land cover types present an area of negative correlations in June which increases with altitude, and likewise, all land cover types present an area of positive correlations in July which decreases with altitude, which is in agreement with previous results obtained in the Carpathians by Sidor et al., 2015, and on areas situated at lower altitudes by Baumbach et al. (2017). Nonetheless, the different land cover types do present different responses to drought conditions, and so the area of negative correlations is always larger on the case of needle leaved forests and smaller for agricultural land, whereas for the case of positive correlations, agricultural land and grassland present the largest areas and needle leaved forests present the smallest.

The drought episode of 2000/2001 was identified as the most severe in the study area in the period 1960-2013 (Spinoni et al., 2013; Ionita et al., 2016), and its spatial coverage reached 60% of Romania (Ionita et al., 2016). Our results also show vegetation under stress conditions on most of the study area, and on roughly 70% of the area these stress conditions lasted 4 or more months. The spatial pattern of vegetation stress obtained with NDVI anomalies is in agreement with the spatial pattern of drought conditions obtained with SPEI, identifying this drought episode as responsible for the reduced vegetation activity. Thus, the relevance of strong drought episodes was clearly illustrated even in humid areas since at higher altitudes, negative NDVI anomalies were persistently low suggesting that although humid vegetation communities show a lower response to climate drought variability, they are also highly sensitive to extreme drought episodes given low ecophysiological resistance to water deficits (Chaves et al., 2003; McDowell et el., 2008)

**Competing interests:** The authors declare that they have no conflict of interest.

**Acknowledgements**

This work was partially supported by national funds through FCT (Fundacão para a Ciência e a Tecnologia, Portugal) under Project IMDROFLOOD (WaterJPI/0004/2014) and the research grant attributed to Ana Russo (SFRH/BPD/99757/2014).

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


| Globcover class label | Label | Area (%) |
|---|---|---|
| Rainfed croplands | Crop | 34.8 |
| Mosaic cropland (50-70%) / natural vegetation (grassland/shrubland/forest) (20-50%) | Crop/Veg | 26.2 |
| Mosaic natural vegetation (grassland/shrubland/forest) (50-70%) / cropland (20-50%) | Veg/Crop | 9.6 |
| Closed (>40%) broad-leaved deciduous forest (>5m) | Broadl dec | 18.2 |
| Closed (>40%) needle-leaved evergreen forest (>5m) | Neddlel everg | 3.3 |
| Closed to open (>15%) mixed broad-leaved and needle-leaved forest (>5m) | Broadl/Neddlel | 2.2 |
| Mosaic grassland (50-70%) / forest or shrubland (20-50%) | Grass | 3.8 |

**Table 1 – Original land cover class labels, land cover label used in this work, and area occupied by each land cover type. Only cover classes covering more than 1% of the territory are shown.**






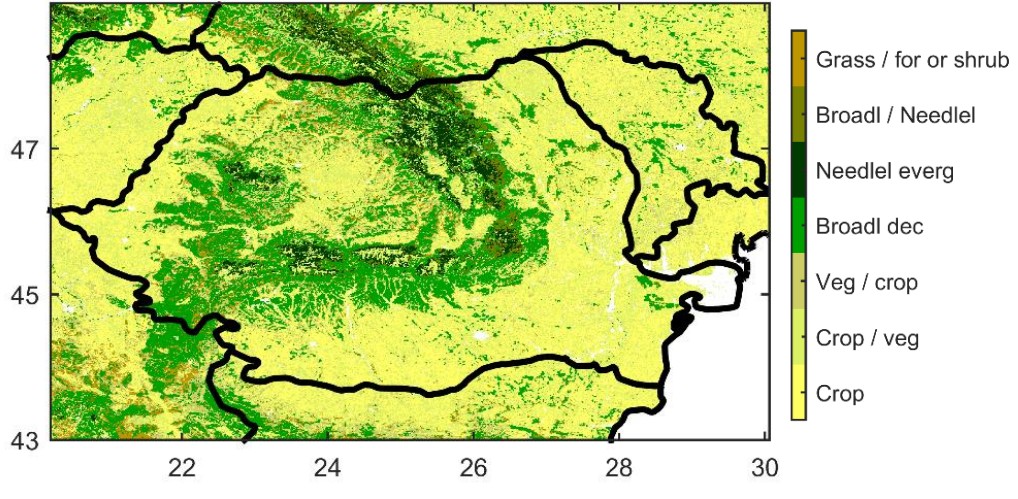

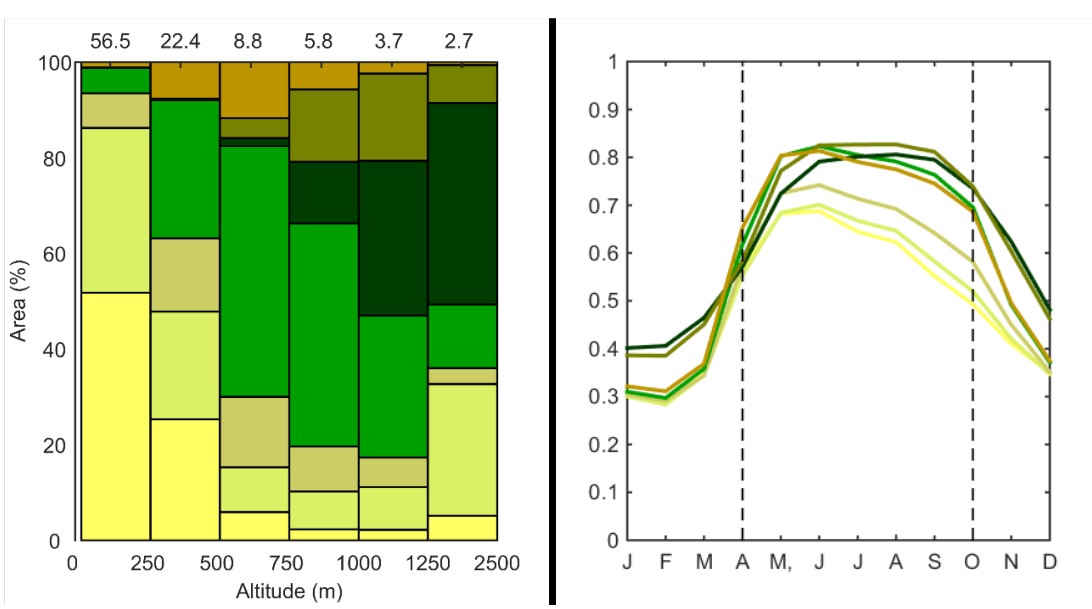

**Figure 1 – Land cover classes, as identified by Globcover 2009 in Southeastern Europe (top panel), area covered by each land cover type along the altitude range (bottom left panel) and median monthly NDVI for each cover class (bottom right panel). Top x-axis on middle panel shows the percentage area of each altitude interval, relative to the total land area.**






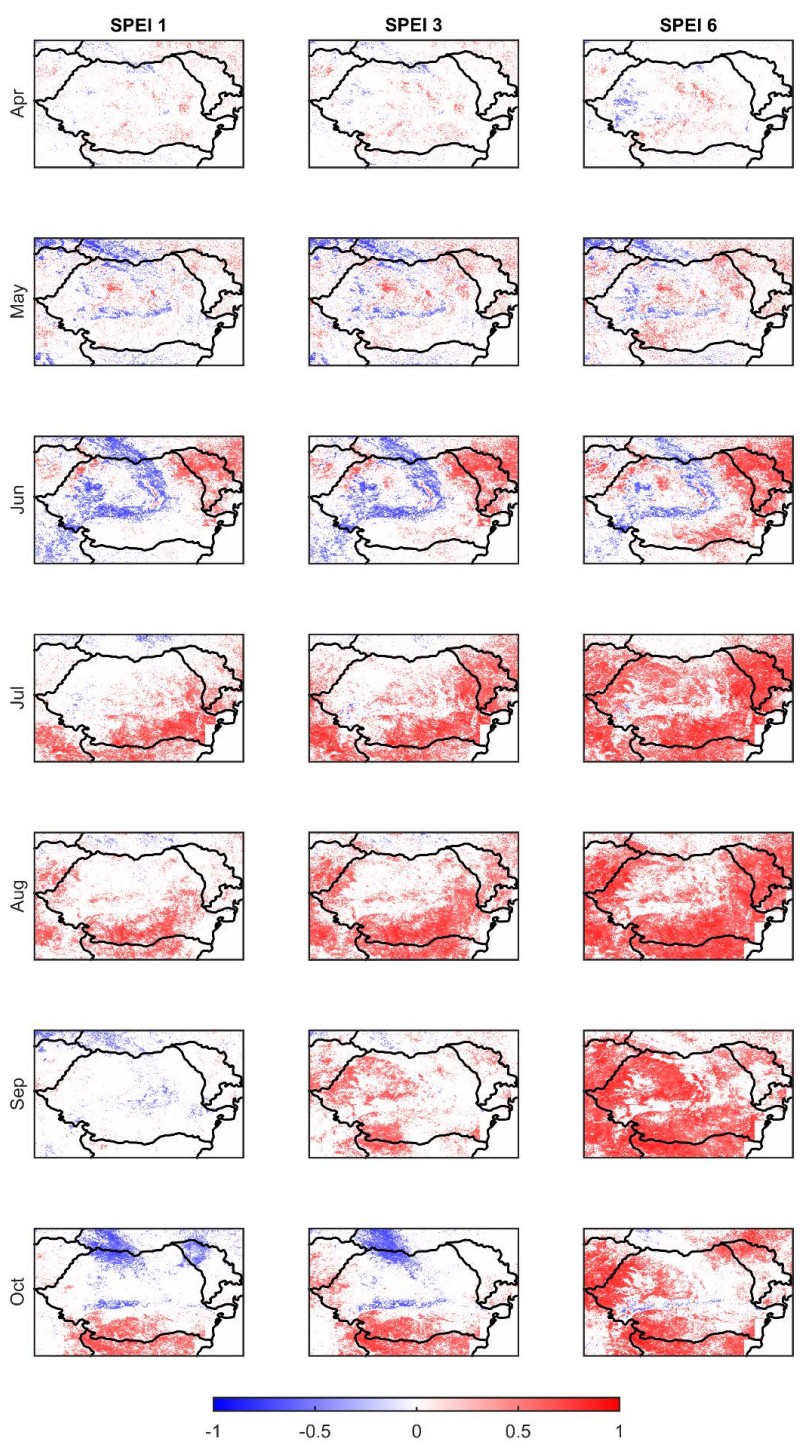

**Figure 2 – Correlation between detrended NDVI and SPEI from April to October for the time scales of 1, 3, 6, months. The minimum significant correlation is ±0.43.**





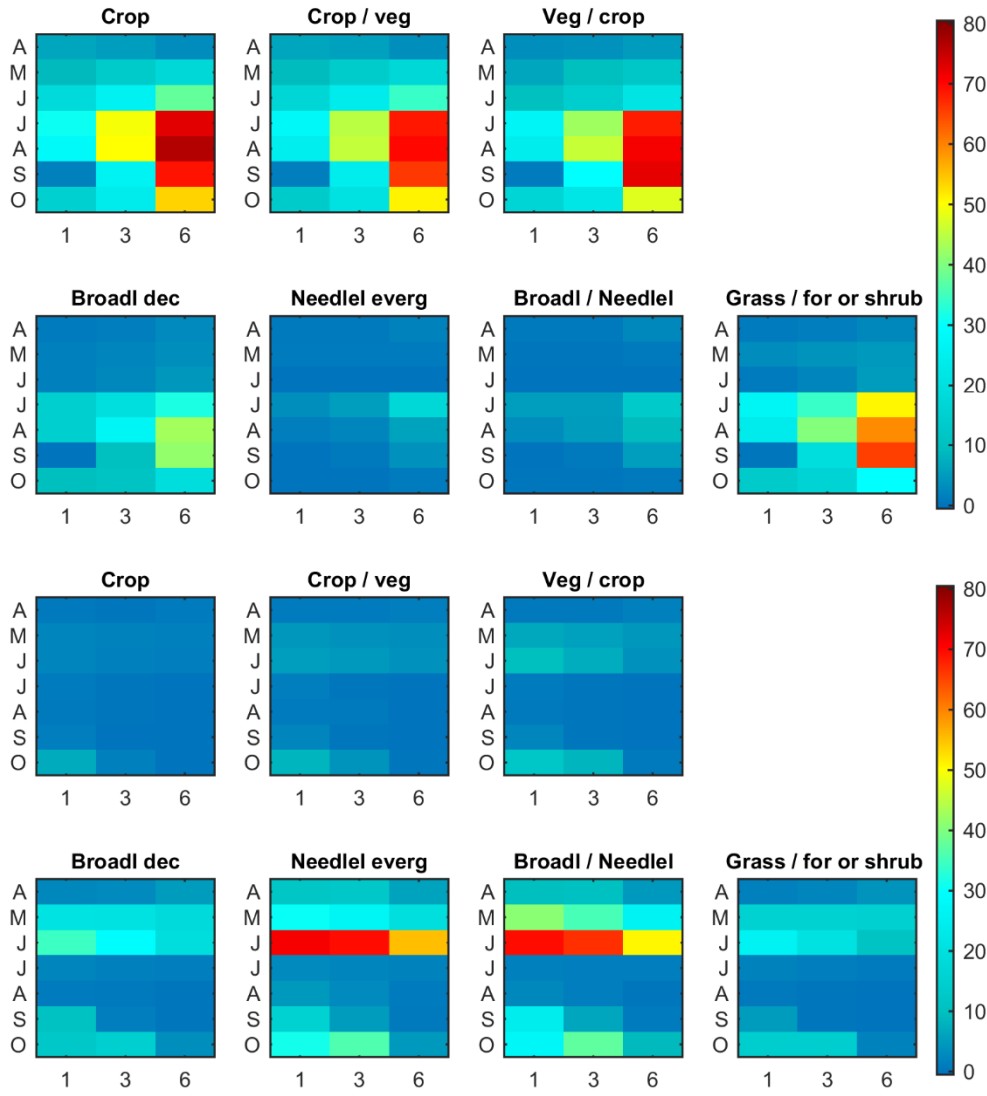

**Figure 3 – Percentage area of each cover class presenting significant positive (top panel) and negative (bottom panel) correlations between NDVI and SPEI, for the months from April to October and time scales of 1, 3, 6, months. Only land cover classes covering more than 1% of the area are shown.**





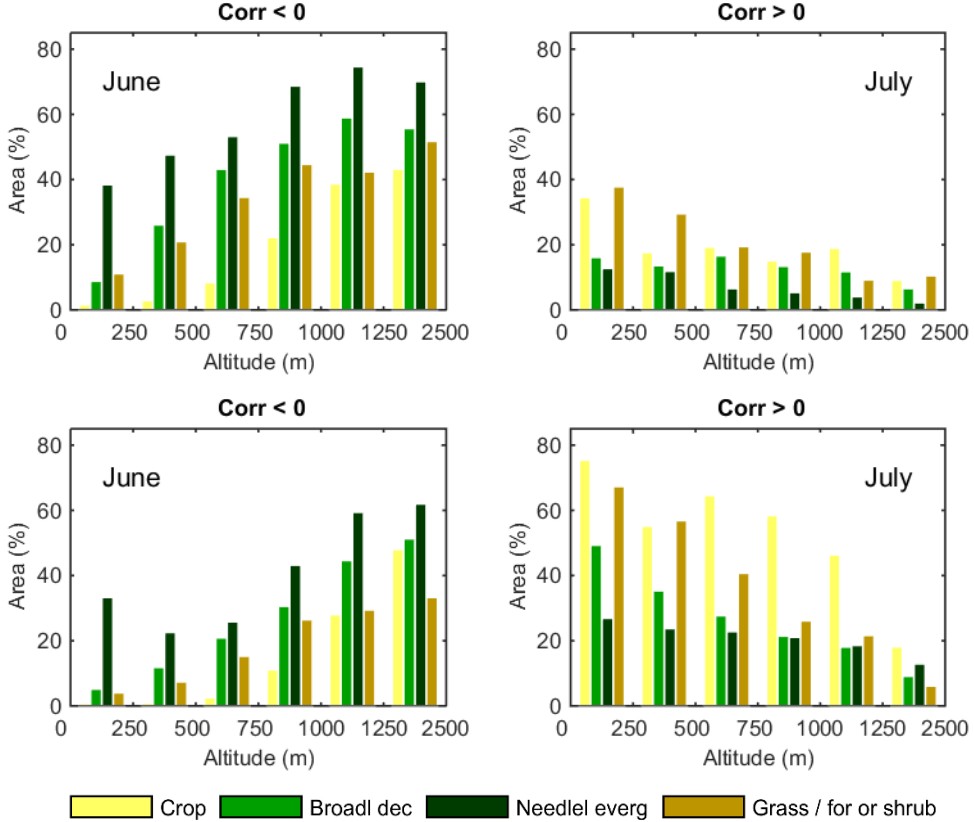


**Figure 4 – Percentage area showing significant correlations in the month of June (left, negative) and July (right, positive), along the altitude range, and for four land cover types, relative to the total area of each land cover type on each altitude interval, for the time scales of 1 (top panel) and 6 months (bottom panel).**





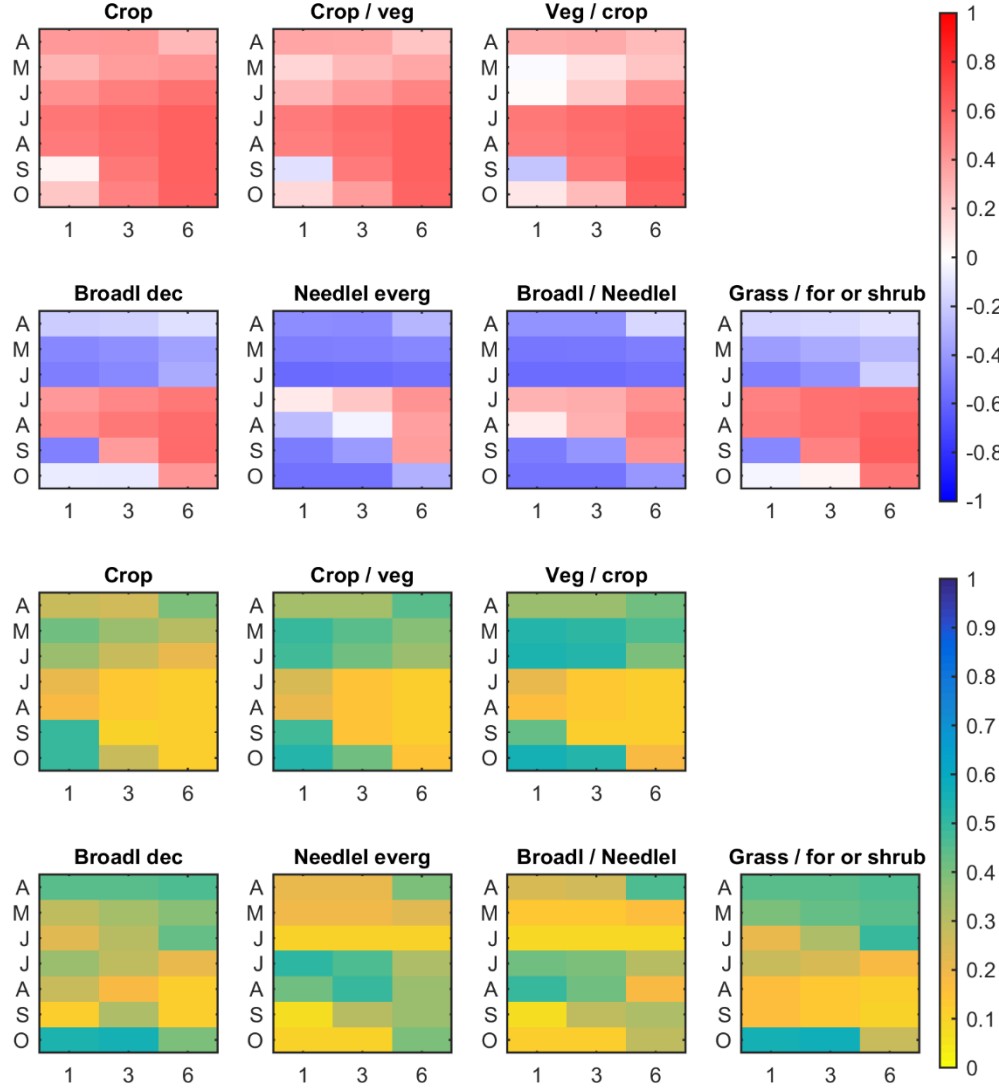

**Figure 5 – Mean value of the significant correlations between detrended NDVI and SPEI (upper panel) and corresponding standard deviation (lower panel).**





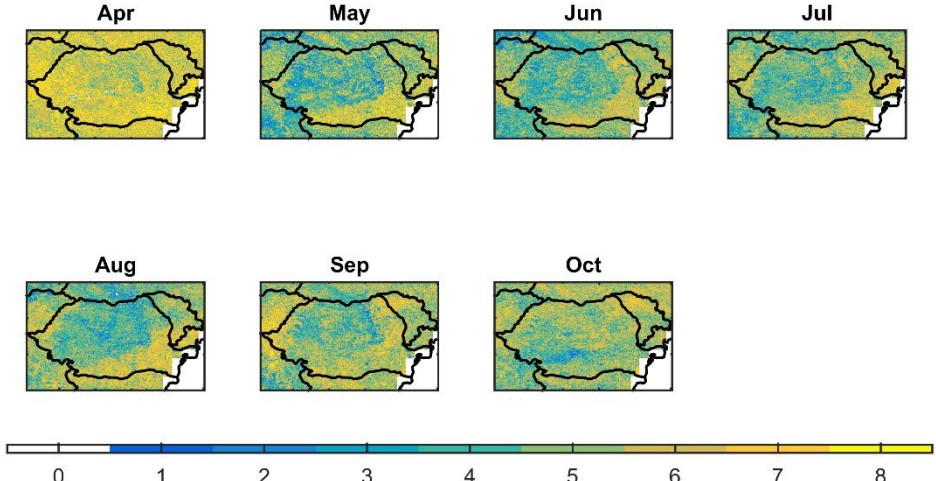

**Figure 6 – Number of NDVI anomalies lower than -0.025, on the entire period, on each month**





**Figure 7 – Simultaneous occurrence of NDVI anomalies lower than -0.025 and SPEI lower than -0.84, shown as a percentage of the number of months showing NDVI anomalies lower than -0.025, for the months of April to October and the time scales of 1, 3, and 6months.**



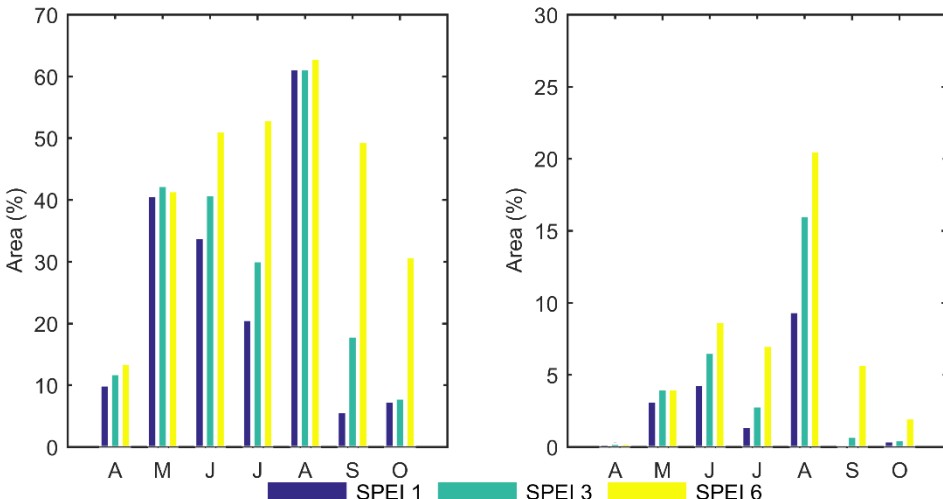

**Figure 8 - Percentage of area showing at least 50% (left panel) and 80% (right panel) of simultaneous occurrence of NDVI anomalies lower than -0.025 and SPEI lower than -0.84.**





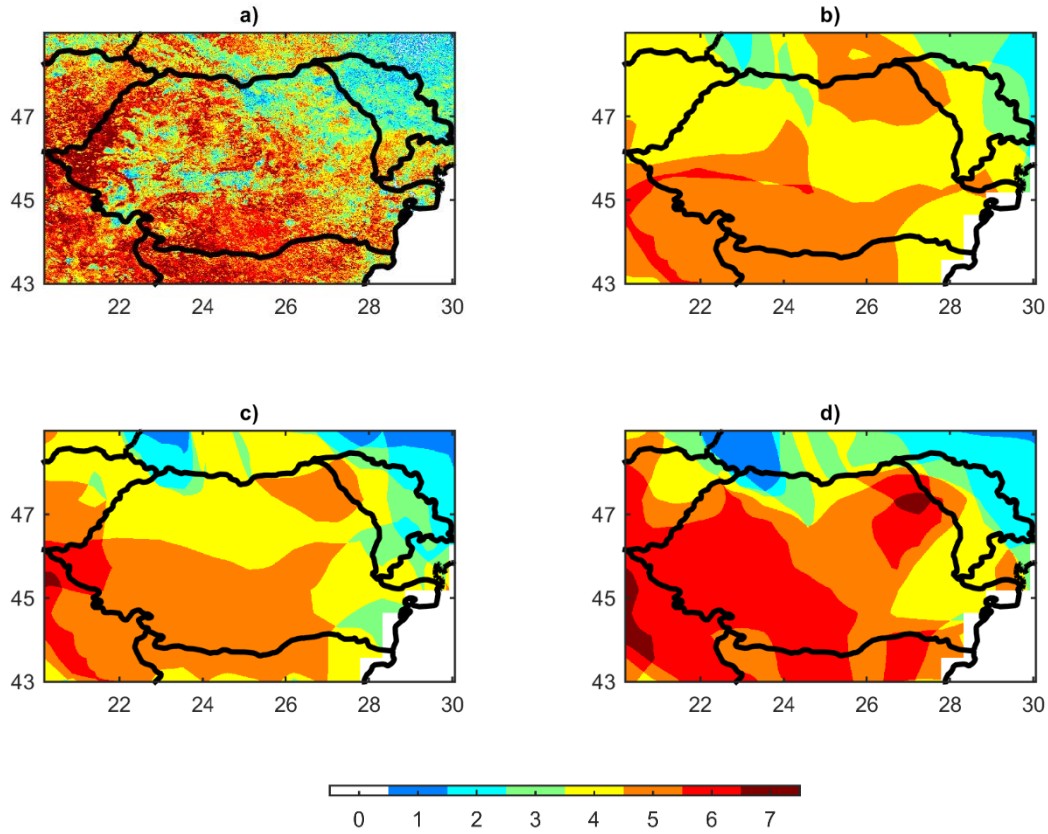

**Figure 9 – a) Persistence of NDVI anomalies below -0.025 and duration of the drought event, in months, as assessed by SPEI with the time scale b) 1, c) 3, and d) 6 months.**



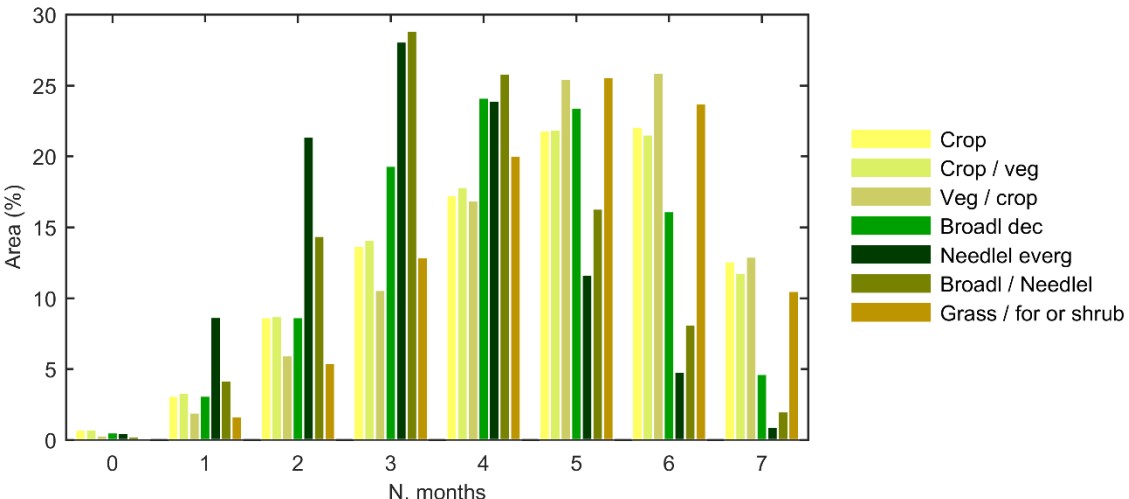

**Figure 10 – Percentage area showing NDVI anomalies below -0.025 between 0 and 7 months on the period April to October 2000, for each cover class**