# Peer review of "Vegetation vulnerability to drought on southeastern Europe"

_Hydrology and Earth System Sciences, 2018_

## Referee Comment (RC1) · Anonymous Referee #1 · 28 Jun 2018

General Comments: 1. Frequency, duration and spatial extent of drought conditions over the study area were considered, but the severity level was not analyzed, which would have been a valuable contribution to this work. Impacts of drought on vegetation vary according to both duration and severity of drought, and the authors did not account for acknowledge this gap in their study. The consideration of severity should would greatly improve this work and the value of the method presented. However, it is acknowledged that this type of research would be complex. If the authors do not choose to add a severity component to the analysis, then at a minimum, they should discuss this gap, state why it was not considered and summarize the limitations of the specific drought conclusions that can be drawn from these results. 2. The paper contains many long sentences that are somewhat difficult to read and interpret the key

points that are trying to be made. Suggest dividing many of the longer sentences into shorter, more precise sentence to improve the readability of the manuscript. Specific Comments: Section 120: What does 'status map' refer to here? Briefly explain what it is. Section 130: State the numerical ranges of the SPEI and corresponding drought severity levels for each range. Section 135: Briefly explain how the SPEI value threshold representing drought was selected from the data for this study. Section 210: The negative correlations and different relationship results for needle-leaved forest is not surprising given their primarily evergreen nature and fairly consistent NDVI response across the growing season. There is minimal NDVI fluctuation across the year because of their evergreen status and minimal NDVI often occur during drought compared to other vegetation types with a more pronounced, seasonal NDVI cycle This may be worth noting somewhere in the paper as this vegetation type appears to have consistently different results in the study. Section 220: What does "intense" refer for "intense negative NDVI…." Do you extremely negative? Consider using a different term here. Section 240 (and throughout paper): Suggest using a different phrase than "period of highest vegetation activity", which is somewhat vague. Possible use a phrase such as "peak of growing season" or "peak of vegetated phase of plants". Section 250: Considerable discussion of the SPEI results for the longer time intervals (6 months+) are presented here and in other parts of the paper, but only minimal discussion of the short time periods (e.g., 1 and 3-month SPEI). Why? This contradicts an earlier statement in the paper where the authors state that the longer period produced similar results and the longest periods would not be presented other than one representative longer time interval, yet the shorter periods are only minimal presented and discussed. Consider adding some discussion about the short SPEI time period results to the paper.

---

## Referee Comment (RC2) · Anonymous Referee #2 · 30 Jun 2018

The title of the manuscript is a mistake, since the content does not refer to other territory than Romania. Eastern Europe holds a very diverse geography, so that doing a study on Romania and pretending it is about Eastern Europe is huge mistake. Such a market strategy is used to attract public and citations, but it is not a fair approach.

Lines 185-190: "Negative correlations are more frequently found on the months of May and June" - can you explain why the correlations are more frequent in May and June? The same amendments would be nice for the next paragraphs, as they are mostly descriptive and not enough explanations are provided. Why no correlation in June for "the area south of the Carpathian Mountains" (L194) and "The areas showing positive correlations from April to June are mostly agricultural land" (L197), so including the areas south of the Carpathians?

[Figure]

Lines 214-219: as mentioned before, the explanations are almost missing, while the very simple description fills the lines. For example, why the standard deviation is lower until July in some forests?

The section 3.3 is a very poor description of the drought event 2000/2001, with no explanation and no positioning in the general context of the droughts in Romania, not to say SE Europe as pretended in the title. The authors claims that "In this section we assess the impact of the strong drought episode of 2000/2001". In reality, the impact has been mentioned very poorly mentioned in the section. This is valid with section 4 too, where the authors claim again that "impacts of droughts on vegetation were analysed". The drought impact is actually indirectly tackled in this paper in the form of the well-known relations between SPEI and NDVI.

---

## Referee Comment (RC3) · Anonymous Referee #3 · 24 Jul 2018

Review for "Vegetation vulnerability to drought on southeastern Europe" by P. Pascoa and co-authors

Summary: The authors present a study that investigates the co-variability of vegetation greenness (based on remotely sensed NDVI) and water availability (assessed through SPEI) over a territory that roughly encompasses the state of Romania. The authors further split up their analysis by altitude and land cover type; and also look at a severe drought episode that occurred in Romania around 2000/2001. While the results are technically sound, and in general a regional-scale analysis of vegetation responses to meteorological variability in different land cover types is interesting, I believe the study could be strongly improved by a suite of potential further in-depth analyses and discussion. The presentation of the study's results appears mainly descriptive, and

a discussion of broader hypotheses, processes or wider implications is missing. An important weakness that needs to be addressed is that correlations are in some cases directly interpreted from a process point of view (e.g. in the Abstract, "[...] a positive effect of dryness on the vegetation activity") - which completely disregards the fact that there are likely confounding meteorological factors that are not analysed. I'll provide a few suggestions below.

Major comments:

* Interpretation of negative correlations between SPEI and NDVI * The authors identify a number of negative correlations between NDVI and SPEI, mostly in early spring and in higher altitudes; and mostly with the shorter time scales of SPEI (e.g. Fig. 5). These are mentioned for instance in the Abstract as one of the highlight results ("a positive effect of dryness on the vegetation activity", p. 1, l. 19/20). This kind of interpretation of a simple correlation analysis is strongly misleading (and the paper relies too much on inferences of this kind): First, correlation does not imply causation (in principle). Second, from a process point of view, it seems more likely that dry springs coincide with warm temperatures / high radiation, which appear a much more likely driver of vegetation activity at high altitude sites (and changes in meteorological drivers from spring to summer and in high altitude sites are not entirely new: See for instance: Jolly et al 2005, GRL; Wolf et al 2016, PNAS; Sippel et al 2017, ERL). This appears also likely, as not only the correlation pattern reverses, but also the relevance of time scales reverses (i.e. in summer it's long SPEI time scales, in spring vegetation responses appear to be on rather short time scales...) While the authors mention that it seems likely that temperature or radiation, and not water availability is limiting vegetation activity in spring, and in higher altitudes (p. 9, l. 279 onwards), a discussion or even further analysis of these likely confounding factors (that are correlated with SPEI!) is missing. Hence, in particular, if the true control of NDVI in spring and at high altitudes would be temperature and/or radiation, and not SPEI, is it meaningful at all to analyse and interpret SPEI/NDVI correlations only? Would it be possible to assess NDVI anoma-

lies jointly, using both SPEI datasets and temperature/radiation datasets? (using for instance partial correlation analysis or other statistical techniques that might be able to account at least to some degree for the covariation of SPEI and temperature/radiation). Also, in light of current literature it would be interesting to analyse whether there are carry-over effects from warm/positive spring conditions towards negative summer conditions? These have been recently found in observations (Buermann et al. 2012 ERL, Wolf et al. 2016) and models (Sippel et al. 2017); and it would be really interesting to analyse whether there is a correlation between spring and summer NDVI conditions? Nonetheless, it would be very interesting to see an analysis that takes into account not only SPEI, but for instance also temperature/radiation in a partial correlation analysis (in which I would guess that the "positive effect of dryness" disappears if controlled for temperature/radiation).

* Descriptive presentations of results * The present text is very hard to read, as it presents numbers over numbers from the analysis, and it appears very descriptive overall. It would probably provide for a better readability if the authors would clearly state 2-3 hypotheses that they investigate, and refer back to these throughout the text. For example, on p. 3, l. 85, the authors state three objectives: "i) to map the vegetation response to drought conditions, ii) to identify the vegetation types that show the strongest response to drought severity and iii) to study the impact of a severe drought episode on the vegetation activity." The readability of the paper might improve if the authors would structure their Discussion&Conclusion Section also along three objectives/hypotheses, and refer back to them in the Discussion&Conclusion section.

* Analysis tools and discussion * The authors use two widely used analysis metrics, namely SPEI and NDVI. However, none of these two metrics is based directly on measurements: SPEI relies on an estimate of the evapotranspiration component; NDVI is remotely sensed vegetation "greenness". While these two metrics are widely used as proxies for dryness and vegetation activity, respectively, it still needs to be emphasized (e.g. in the methods section) that both metrics are only highly simplified proxies of

dryness/vegetation activity; and thus that some care needs to be taken in interpretation of results due to these methodological choices. In a similar vein, it would be very interesting to see a discussion on how the results of this paper compare to analyses of direct vegetation growth measurements (e.g. FLUXNET) in response to meteorological variability. There are some papers available, some that analyse also stratified by land cover type /vegetation type: e.g. Schwalm et al. 2010 GCB; von Buttlar et al. 2018 BG.

Minor comments: ————————————————————

p. 1, title: shouldn't is say "vegetation vulnerability to drought IN southeastern Europe" p. 1, l. 17 "on July and August" should read "in July and August". Several more writing issues across the text. p. 4, Section 2.2 on NDVI data. Are NDVI time series deseasonalized? This would be reasonable for looking at vegetation activity anomalies. Is there any pre-processing done on NDVI values other than discarding values below 0.1. p. 5, l. 136. A threshold of -0.84 appears very subjective. Could you maybe make a plot to show where this threshold lies across the SPEI values obtained in the study? p. 5, line 150: What about sensor ageing? p. 6 line 187: should read "shown IN fig. 3"

Fig. 4: "Significant correlation" (figure caption) and "Corr > 0" (title of figure) must be a diffent thing.

References: ————————————————— Buermann, W., Bikash, P.R., Jung, M., Burn, D.H. and Reichstein, M., 2013. Earlier springs decrease peak summer productivity in North American boreal forests. Environmental Research Letters, 8(2), p.024027.

Jolly, W.M., Dobbertin, M., Zimmermann, N.E. and Reichstein, M., 2005. Divergent vegetation growth responses to the 2003 heat wave in the Swiss Alps. Geophysical Research Letters, 32(18).

Schwalm, C.R., Williams, C.A., Schaefer, K., Arneth, A., Bonal, D., Buchmann, N., Chen, J., Law, B.E., Lindroth, A., Luyssaert, S. and Reichstein, M., 2010. Assimilation exceeds respiration sensitivity to drought: A FLUXNET synthesis. Global Change Biology, 16(2), pp.657-670.

Sippel, S., Forkel, M., Rammig, A., Thonicke, K., Flach, M., Heimann, M., Otto, F.E., Reichstein, M. and Mahecha, M.D., 2017. Contrasting and interacting changes in simulated spring and summer carbon cycle extremes in European ecosystems. Environmental Research Letters, 12(7), p.075006.

von Buttlar, Jannis, et al. "Impacts of droughts and extreme-temperature events on gross primary production and ecosystem respiration: a systematic assessment across ecosystems and climate zones." Biogeosciences 15.1 (2018): 1293-1318.

Wolf, S., Keenan, T.F., Fisher, J.B., Baldocchi, D.D., Desai, A.R., Richardson, A.D., Scott, R.L., Law, B.E., Litvak, M.E., Brunsell, N.A. and Peters, W., 2016. Warm spring reduced carbon cycle impact of the 2012 US summer drought. Proceedings of the National Academy of Sciences, 113(21), pp.5880-5885.

---

## Author Comment (AC1) · 4 Sep 2018

1. Frequency, duration and spatial extent of drought conditions over the study area were considered, but the severity level was not analyzed, which would have been a valuable contribution to this work. Impacts of drought on vegetation vary according to both duration and severity of drought, and the authors did not account for acknowledge this gap in their study. The consideration of severity should would greatly improve this work and the value of the method presented. However, it is acknowledged that this type of research would be complex. If the authors do not choose to add a severity component to the analysis, then at a minimum, they should discuss this gap, state why it was not considered and summarize the limitations of the specific drought conclusions that can be drawn from these results.

[Figure]

Response: We acknowledge that the level of severity was not assessed separately, however the effect of severity is indirectly included. The existence of a linear relation between NDVI and SPEI, as assessed by the correlation analysis, points to the simultaneous occurrence of extreme NDVI anomalies and SPEI values. Therefore, for the cases when the correlation is positive, we expect that NDVI anomalies and SPEI values decrease simultaneously. Moreover, it should be noted that we also included the analysis of a severe drought event to study in more detail the impacts on vegetation. In fact, the drought event of 2000/2001 was chosen precisely because it was found to be the most severe in the period 1960-2013, as stated by several authors (Spinoni et al., 2013, Ionita et al. 2016). The study of this drought event showed that severe droughts affect vegetation in areas usually not affected by droughts. This is already stated in the data and methods section.

An analysis of the effect of different classes of severity should be performed on a pixel basis, in order to discard differences arising from different geographical locations. We believe this analysis would require longer time series in order to capture drought events of different severity classes on the same location (pixel). The NDVI dataset used in this work is not long enough to perform such analysis. This statement will be included in the manuscript.

2. The paper contains many long sentences that are somewhat difficult to read and interpret the key points that are trying to be made. Suggest dividing many of the longer sentences into shorter, more precise sentence to improve the readability of the manuscript.

Response: We have made a serious effort to accommodate this requirement. Therefore, several longer sentences will be split in two shorter sentences in the final version of the paper, as follows:

Line 185: The number of months with SPEI values below -0.84 was also computed and mapped for the same time period. This allowed assessing the spatial extent of the

drought episode at various time scales and the overlapping of vegetation under stress conditions.

Line 234: The Carpathian Mountains generally present a lower concordance than the remaining area, and it also presents a lower number of low NDVI anomalies (Fig.6). A higher agreement is found in the central area in May, in the northeastern area in June and in the south in July (Fig.7).

Line 298: By analysing the period from May to October, it was possible to identify significant negative correlations at high altitudes in the Carpathians, and also the moment these negative correlations tend to disappear, around in July. This coincides with the emergence of significant positive correlations at lower altitudes.

Line 304: The negative correlations obtained here for the month of June at high altitudes point to a positive effect of dryness on the vegetation activity. The subsequent signal change in the month of July may be a response to the increasing temperature coupled to the decrease in precipitation, and an indicator of water stress, even though at shorter time scales the correlation at high altitudes is mostly non-significant.

Line 338: In any case, all land cover types present an area of negative correlations in June which increases with altitude, and likewise, all land cover types present an area of positive correlations in July which decreases with altitude. These results are in agreement with previous results obtained in the Carpathians by Sidor et al., 2015, and on areas situated at lower altitudes by Baumbach et al. (2017).

Section 120: What does 'status map' refer to here? Briefly explain what it is.

Response: This dataset provides information regarding the presence of clouds, cloud shadows, or ice, at each pixel, which is referred to as the status map. Pixels identified as snow cover by the status map were excluded, as well as NDVI values below 0.1, since they do not represent vegetation. A brief description of the status map will be included in the manuscript and the last sentence of the section will be changed in order

to make more clear that NDVI values below 0.1 are unrelated to the status map. Section 130: State the numerical ranges of the SPEI and corresponding drought severity levels for each range.

Response: The severity classification used in this work was proposed by Agnew (2000). The limits and probability of occurrence of each class will be included in the new version of the manuscript, on Table 2. Theoretically, SPEI has no limits, although the probability of occurrence of extreme values is extremely rare, since this index is normally distributed. The information attached will be added in order to clarify this aspect and based on this classification, a threshold of -0.84 was established to identify the drought events (20% of the probability distribution function and a return period of 1 in 5 years).

Section 135: Briefly explain how the SPEI value threshold representing drought was selected from the data for this study.

Response: As mentioned in the previous comment, the threshold was chosen based on the normal distribution function. The meteorological data does not influence the choice of the thresholds, since the resulting index is standardized and therefore these thresholds always correspond to the same probability of occurrence.

Section 210: The negative correlations and different relationship results for needle-leaved forest is not surprising given their primarily evergreen nature and fairly consistent NDVI response across the growing season. There is minimal NDVI fluctuation across the year because of their evergreen status and minimal NDVI often occur during drought compared to other vegetation types with a more pronounced, seasonal NDVI cycle This may be worth noting somewhere in the paper as this vegetation type appears to have consistently different results in the study.

Response: We agree with reviewer about the less marked vegetative cycle of NDVI over needle-leaved forest. Several works pointed out the different response of this vegetation type to drought events (Gouveia et al., 2009, 2012, 2017, Vicente-Serrano

et al., 2013). The knowledge about this behaviour lead us to compute the annual cycle of NDVI for the different land cover types over the region (Figure 1, bottom right panel). Additionally, we opted to perform the correlation analysis on monthly NDVI, as our aim is to analyse the inter-annual variability of NDVI and therefore assess the inter-annual response of vegetation to extreme events, namely drought events.

However, and with the aim of clarifying this issue a reference to the less marked vegetative cycle of needle-leaved will be added on the Section Discussion and Conclusions in the next version of the paper.

Section 220: What does "intense" refer for "intense negative NDVI: : :." Do you extremely negative? Consider using a different term here.

Response: The expression will be changed in order to clarify this issue as follows:

'The relationship between drought severity and the NDVI was also assessed by counting the number of months characterised by simultaneous occurrence of extremely negative NDVI anomalies and negative SPEI.'

The thresholds used to consider the NDVI anomaly as extremely negative is -0.025 and this threshold is clearly mentioned on the Data and Methods Section.

Section 240 (and throughout paper): Suggest using a different phrase than "period of highest vegetation activity", which is somewhat vague. Possible use a phrase such as "peak of growing season" or "peak of vegetated phase of plants".

Response: We acknowledge the reviewer's point of view. Throughout the text, the expression 'highest vegetation activity' does not corresponds to a single month but refers to the period in which the vegetation is photosynthetically more active and therefore prone to be more affected by adverse conditions. Therefore, we agree with the reviewer that a more appropriate terminology should be used ("growing season") and several changes will be made on the new version of the document.

Section 250: Considerable discussion of the SPEI results for the longer time intervals

(6 months+) are presented here and in other parts of the paper, but only minimal discussion of the short time periods (e.g., 1 and 3-month SPEI). Why? This contradicts an earlier statement in the paper where the authors state that the longer period produced similar results and the longest periods would not be presented other than one representative longer time interval, yet the shorter periods are only minimal presented and discussed. Consider adding some discussion about the short SPEI time period results to the paper.

Response: We understand the reviewer's concern, as in the first version of this work we decided to discuss the results only in the last section of the manuscript (on Discussion and Conclusions section). On the other hand, our statement about the results related with time scales of 9 and 12 months aims to highlight that these results do not add additional information to the results obtained with the time scale of 6 months.

Nevertheless, we agree with the reviewer that the discussion of the results should indeed include more details about the shorter time scales (1 to 3 months). Therefore, we will add a short statement about this issue on Results and more detailed information on Discussion and Conclusions section.

References: Agnew, C.T., Using the SPI to identify drought. Drought Network News, vol. 12, no. 1, pp. 6–12, 2000. Gouveia, C., Trigo, R.M., DaCamara, C.C.: Drought and vegetation stress monitoring in Portugal using satellite data, Nat. Hazard. Earth. Sys., 9, 185-195, 2009, https://doi.org/10.5194/nhess-9-185-2009

Gouveia, C.M., Bastos, A., Trigo, R.M., DaCamara, C.C.: Drought impacts on vegetation in the pre- and post-fire events over Iberian Peninsula, Nat. Hazard. Earth. Sys., 12, 3123-3137, doi:10.5194/nhess-12-3123-2012, 2012.

Gouveia, C.M., Trigo, R.M., Beguería, S., Vicente-Serrano, S.M.: Drought impacts on vegetation activity in the Mediterranean region: An assessment using remote sensing data and multi-scale drought indicators, Global Planet. Chan., 151: 15-27, 2017. https://doi.org/10.1016/j.gloplacha.2016.06.011.

Vicente-Serrano, S.M., Gouveia, C., Camarero, J.J., Beguería, S., Trigo, R., López-Moreno, J.I,m Azorín-Molina, C., Pasho, E., Lorenzo-Lacruz, J., Revuelto, J., Morán-Tejeda, E., Sanchez-Lorenzo, A.: Response of vegetation to drought timescales across global land biomes. P. Natl. Acad. Sci., 110(1), 52-57, https://doi.org/10.1073/pnas.1207068110, 2013.
* * *
[Figure]

| Drought classification | SPEI range | Return period |
|---|---|---|
| Moderate | -0.84<SPEI | 1 in 5 years |
| Severe | -1.28<SPEI | 1 in 10 years |
| Extreme | -1.65<SPEI | 1 in 20 years |

**Table – Drought classes and corresponding range of SPEI values and return period, as proposed by Agnew (2000) and used in this work.**

**Fig. 1.** Drought classes and corresponding range of SPEI values and return period, as proposed by Agnew (2000) and used in this work.

---

## Author Comment (AC2) · 4 Sep 2018

Summary: The authors present a study that investigates the co-variability of vegetation greenness (based on remotely sensed NDVI) and water availability (assessed through SPEI) over a territory that roughly encompasses the state of Romania. The authors further split up their analysis by altitude and land cover type; and also look at a severe drought episode that occurred in Romania around 2000/2001. While the results are technically sound, and in general a regional-scale analysis of vegetation responses to meteorological variability in different land cover types is interesting, I believe the study could be strongly improved by a suite of potential further in-depth analyses and discussion. The presentation of the study's results appears mainly descriptive, and a discussion of broader hypotheses, processes or wider implications is missing. An

important weakness that needs to be addressed is that correlations are in some cases directly interpreted from a process point of view (e.g. in the Abstract, "[...] a positive effect of dryness on the vegetation activity") - which completely disregards the fact that there are likely confounding meteorological factors that are not analysed. I'll provide a few suggestions below.

Response: We appreciate the overall good appreciation of our manuscript by the reviewer. The issues addressed by the reviewer are commented below.

Major comments: 1. * Interpretation of negative correlations between SPEI and NDVI * The authors identify a number of negative correlations between NDVI and SPEI, mostly in early spring and in higher altitudes; and mostly with the shorter time scales of SPEI (e.g. Fig. 5). These are mentioned for instance in the Abstract as one of the highlight results ("a positive effect of dryness on the vegetation activity", p. 1, l. 19/20). This kind of interpretation of a simple correlation analysis is strongly misleading (and the paper relies too much on inferences of this kind): First, correlation does not imply causation (in principle).

Response: We appreciate the reviewer's concerns regarding the interpretation of the results obtained with the correlation analysis. Firstly, we would like to stress that we did not intend to overemphasise the existence of a causation relation based on these results. Furthermore, we intent to write carefully aiming to avoid misleading interpretations about those relationships. When we mentioned 'the positive effect of dryness on vegetation activity' we were pointing to the relationship observed, by means of correlation analysis, when years with low vegetation activity are coincident with years of negative SPEI values (drought conditions). This behaviour is similar to the ones obtained in several other works for several other regions in the world using the same methodology (Ji and Peters, 2003; Lotsch et al., 2003; 2005; Quiring et al., 2010; Brown et al., 2008; Wang et al., 2015; Vicente Serrano et., 2012; 2013; 2014; Gouveia et al., 2017). However, we have included other methodologies, namely the one proposed by Gouveia et al., (2009) and applied over different regions (e.g. Fertile Crescent - Trigo at al., 2010;

China - Barriopedro et al., 2012, Madeira - Liberato et al., 2017) that together help understand the effect of these variables.

2. Second, from a process point of view, it seems more likely that dry springs coincide with warm temperatures / high radiation, which appear a much more likely driver of vegetation activity at high altitude sites (and changes in meteorological drivers from spring to summer and in high altitude sites are not entirely new: See for instance: Jolly et al 2005, GRL; Wolf et al 2016, PNAS; Sippel et al 2017, ERL).

Response: We thank the reviewer for the suggested papers. One of our main goals was to map the vegetation response in the area to drought conditions, and although we did not intend to focus on the altitude gradient, this issue had to be explored, since it is intimately related with the maps obtained. We did not think the results we obtained regarding the altitude range were a scientific novelty, and we have cited in the text several authors that obtained comparable or similar results in the study area (Gouveia et al., 2017, Sidor et al. (2015)). However we acknowledge the relevance of the driving mechanism suggested by the reviewer, as well as the additional references suggested by the reviewer and will therefore also include in our discussion the points indicated by the reviewer.

3. This appears also likely, as not only the correlation pattern reverses, but also the relevance of time scales reverses (i.e. in summer it's long SPEI time scales, in spring vegetation responses appear to be on rather short time scales...) While the authors mention that it seems likely that temperature or radiation, and not water availability is limiting vegetation activity in spring, and in higher altitudes (p. 9, l. 279 onwards), a discussion or even further analysis of these likely confounding factors (that are correlated with SPEI!) is missing. Hence, in particular, if the true control of NDVI in spring and at high altitudes would be temperature and/or radiation, and not SPEI, is it meaningful at all to analyse and interpret SPEI/NDVI correlations only?

Response: We understand the reviewer's point, however we should stress that the

aim of our work is to analyse and characterize the relationship between drought and vegetation activity using a multiscalar drought index and vegetation indices obtained by satellite. The processes behind these relationships are out of the scope of this work. As we recognize that radiation and temperature are important factors that should be included in drought impacts, we opted by using the SPEI. SPEI incorporates a simple water balance, and if temperature increases vegetation activity, it may affect the available water, thus affecting the vegetation activity. For this reason, we consider that the analysis of NDVI/SPEI correlation may provide important information.

Moreover, and accordingly with the above mentioned goals, we did mention that it was likely that temperature or radiation was a limiting factor in the high altitudes, based on other works that have shown this (Nemani et al., 2003). Nonetheless, we cannot rule out the influence of precipitation in the vegetation activity, particularly since other works have shown that precipitation does have a correlation with vegetation activity in the study area, at high altitudes (Levanič et al. (2013), Sidor et al. (2015)). We have mentioned this in the Discussion and Conclusion section (please see page 9, lines 272- 279 and page 10, lines 293-296

Although recognizing the great interest in analysing radiation and temperature and also SPEI and NDVI over high altitudes is not the goal of this work, as it is not possible to do this kind of analysis with gridded data with 0.5° of spatial resolution. Nonetheless, we will improve the Discussion and Conclusion section, in order to include other effects of these meteorological variables, that may not be very clear in the original manuscript, as well as the role of the SPEI time scale.

4. Would it be possible to assess NDVI anomalies jointly, using both SPEI datasets and temperature/radiation datasets? (using for instance partial correlation analysis or other statistical techniques that might be able to account at least to some degree for the covariation of SPEI and temperature/radiation).

Response: This analysis would certainly provide important information, but, due to the

reasons presented above, we consider it is outside the scope of this work, which is to assess the influence of drought on vegetation in the study area. The analysis proposed by the reviewer will enlarge the paper and defocus the manuscript.

5. Also, in light of current literature it would be interesting to analyse whether there are carry-over effects from warm/positive spring conditions towards negative summer conditions? These have been recently found in observations (Buermann et al. 2012 ERL, Wolf et al. 2016) and models (Sippel et al. 2017); and it would be really interesting to analyse whether there is a correlation between spring and summer NDVI conditions?

Response: The effect of past climate conditions is included in the different SPEI time scales considered in this work, although the results were not discussed considering the proposed point of view. We do consider it is an interesting point and will include it in the discussion. It is possible that pixels showing both positive correlations in July and negative correlations in June are reflecting the referred carry-over effect.

An in-depth analysis of the carry-over effects mentioned by the reviewer would need a longer time series, in order to include the different climate conditions on both seasons on a pixel basis. It is likely that the dataset used in this work is not long enough to this type of analysis. However a discussion of these aspects will be included in Discussion, mentioning the works referred by the reviewer.

6. Nonetheless, it would be very interesting to see an analysis that takes into account not only SPEI, but for instance also temperature/radiation in a partial correlation analysis (in which I would guess that the "positive effect of dryness" disappears if controlled for temperature/radiation).

Response: Please consider the response to comment 4.

7. * Descriptive presentations of results * The present text is very hard to read, as it presents numbers over numbers from the analysis, and it appears very descriptive overall. It would probably provide for a better readability if the authors would clearly

state 2-3 hypotheses that they investigate, and refer back to these throughout the text. For example, on p. 3, l. 85, the authors state three objectives: "i) to map the vegetation response to drought conditions, ii) to identify the vegetation types that show the strongest response to drought severity and iii) to study the impact of a severe drought episode on the vegetation activity." The readability of the paper might improve if the authors would structure their Discussion&Conclusion Section also along three objectives/ hypotheses, and refer back to them in the Discussion&Conclusion section.

Response: We thank the suggestion to improve the readability of our work. We initially decided to discuss the results only in the last section, but taking into account the comments from all the reviewers we will include in the Results section some interpretation, so it won't be as descriptive. Accordingly, we will also improve the Discussion section, as suggested.

8. * Analysis tools and discussion * The authors use two widely used analysis metrics, namely SPEI and NDVI. However, none of these two metrics is based directly on measurements: SPEI relies on an estimate of the evapotranspiration component; NDVI is remotely sensed vegetation "greenness". While these two metrics are widely used as proxies for dryness and vegetation activity, respectively, it still needs to be emphasized (e.g. in the methods section) that both metrics are only highly simplified proxies of dryness/vegetation activity; and thus that some care needs to be taken in interpretation of results due to these methodological choices.

Response: We agree with the precautionary note raised by the Reviewer. We will include in the Data and Methods Section more information regarding the indices, such as their advantages and caveats. We would like to stress that we have included in the Introduction information that justifies the use of these indices, such as the difficulty to directly quantify droughts (thus the need to use a proxy).

9. In a similar vein, it would be very interesting to see a discussion on how the results of this paper compare to analyses of direct vegetation growth measurements (e.g.

Interactive
comment

FLUXNET) in response to meteorological variability. There are some papers available, some that analyse also stratified by land cover type /vegetation type: e.g. Schwalm et al. 2010 GCB; von Buttlar et al. 2018 BG.

Response: We have cited in the Discussion Section several authors that used tree-ring data and crop yield data to assess the impact of drought/climate variability (Levanič et al. (2013), Sidor et al. (2015), Potopová et al., 2016). Unfortunately, we have not found works by other authors using FLUXNET data in this region that we could use to compare. Nonetheless, it is our opinion that the methodology used is robust, which is reinforced by the similarity with the results obtained by the mentioned works. A reference to the mentioned papers will also be included in Discussion.

Minor comments: —————————————— 10. p. 1, title: shouldn't is say "vegetation vulnerability to drought IN southeastern Europe"

Response: We thank the reviewer for the correction.

11. p. 1, l. 17 "on July and August" should read "in July and August". Several more writing issues across the text.

Response: We thank the reviewer for the corrections. We will perform a thorough review of the manuscript in order to correct other issues.

12. p. 4, Section 2.2 on NDVI data. Are NDVI time series deseasonalized? This would be reasonable for looking at vegetation activity anomalies. Is there any pre-processing done on NDVI values other than discarding values below 0.1.

Response: The monthly time series of NDVI were detrended. Detrending is a common procedure when performing a correlation analysis. On the other hand, since the de-trending was performed on the monthly values, the resulting anomaly is in reference to the monthly values and therefore we do not see the need to deseasonalize the time series.

13. p. 5, l. 136. A threshold of -0.84 appears very subjective. Could you maybe make

a plot to show where this threshold lies across the SPEI values obtained in the study?

Response: As mentioned in the Introduction, one of the reasons making it so difficult to directly quantify drought is the difficulty to precisely identify the onset and end of a drought event. Bearing this in mind, the choice of a threshold in this situation is inherently subjective. We have chosen this threshold based on the probability distribution function of the SPEI. Really, any threshold is always subjective, but given different vulnerabilities and responses of natural vegetation to droughts, it is really difficult to establish objective thresholds. Considering the threshold -0.84, the frequency of SPEI values indicating drought (moderate, severe or extreme) is around 35% in the entire period used to compute the index. We show on Fig.1 the requested plot, for the time scales considered in the study, but we do not think necessary to include in the manuscript.

14. p. 5, line 150: What about sensor ageing?

Response: The dataset used is the Collection 3, which is the result of the reprocessing of Collections 1 and 2, and Toté et al. (2017) concluded that this version is more stable over time when compared with the previous dataset. Changes in illumination associated with sensor drift may introduce some artificial trends in the time series, but if occurring, these were eliminated during the detrending of the time series. No other trend analysis was performed in this work that could be masked by changes associated with sensor drift.

15. p. 6 line 187: should read "shown IN fig. 3"

Response: We thank the reviewer for the correction.

16. Fig. 4: "Significant correlation" (figure caption) and "Corr > 0" (title of figure) must be a diffent thing.

Response: The figure caption will be clarified, in order to clarify this issue. It will read: "Percentage area showing significant correlations in the month of June (left, negative

correlation) and July (right, positive correlation)."

References: Buermann, W., Bikash, P.R., Jung, M., Burn, D.H. and Reichstein, M., 2013. Earlier springs decrease peak summer productivity in North American boreal forests. Environmental Research Letters, 8(2), p.024027. Jolly, W.M., Dobbertin, M., Zimmermann, N.E. and Reichstein, M., 2005. Divergent vegetation growth responses to the 2003 heat wave in the Swiss Alps. Geophysical Research Letters, 32(18). Schwalm, C.R., Williams, C.A., Schaefer, K., Arneth, A., Bonal, D., Buchmann, N., Chen, J., Law, B.E., Lindroth, A., Luyssaert, S. and Reichstein, M., 2010. Assimilation exceeds respiration sensitivity to drought: A FLUXNET synthesis. Global Change Biology, 16(2), pp.657-670. Sippel, S., Forkel, M., Rammig, A., Thonicke, K., Flach, M., Heimann, M., Otto, F.E., Reichstein, M. and Mahecha, M.D., 2017. Contrasting and interacting changes in simulated spring and summer carbon cycle extremes in European ecosystems. Environmental Research Letters, 12(7), p.075006. von Buttlar, Jannis, et al. "Impacts of droughts and extreme-temperature events on gross primary production and ecosystem respiration: a systematic assessment across ecosystems and climate zones." Biogeosciences 15.1 (2018): 1293-1318. Wolf, S., Keenan, T.F., Fisher, J.B., Baldocchi, D.D., Desai, A.R., Richardson, A.D., Scott, R.L., Law, B.E., Litvak, M.E., Brunsell, N.A. and Peters, W., 2016. Warm spring reduced carbon cycle impact of the 2012 US summer drought. Proceedings of the National Academy of Sciences, 113(21), pp.5880-5885.

References:

Barriopedro D., Gouveia C.M., Trigo R.M., Wang L., The 2009/10 Drought in China: Possible Causes and Impacts on Vegetation. Jounral fo Hydrometeorology, Vol 13., 2012 DOI: 10.1175/JHM-D-11-074.1 Brown, J.F., Wardlow, B.D., Tadesse, T., Hayes, M., J., Reed, B.C., The Vegetation Drought Response Index (VegDRI): A New Integrated Approach for Monitoring Drought Stress in Vegetation, GIScience & Remote Sensing, 45(1), 16-46, 2008

Gouveia, C., Trigo, R.M., DaCamara, C.C.: Drought and vegetation stress monitoring in Portugal using satellite data, Nat. Hazard. Earth. Sys., 9, 185-195, https://doi.org/10.5194/nhess-9-185-2009, 2009

Gouveia, C.M., Trigo, R.M., Beguería, S., Vicente-Serrano, S.M.: Drought impacts on vegetation activity in the Mediterranean region: An assessment using remote sensing data and multi-scale drought indicators, Global Planet. Chan., 151: 15-27. https://doi.org/10.1016/j.gloplacha.2016.06.011, 2017.

Ji, L., Peters, A.J., Assessing vegetation response to drought in the northern Great Plains using vegetation and drought indices, Remote Sensing of Environment, 87(1), 85-98, 2003, https://doi.org/10.1016/S0034-4257(03)00174-3.

Levanič, T., Popa, I., Poljanšek, S., Nechita, C.: A 323-year long reconstruction of drought for SW Romania based on black pine (Pinus Nigra) tree-ring widths, Int. J. Biometeorol., 57, 703-714, doi: 10.1007/s00484-012-0596-9, 2013.

Liberato M. L. R., Ramos A. M., Gouveia C. M., Sousa P., Russo A., Trigo R.M., Santo F.E., Exceptionally extreme drought in Madeira Archipelago in 2012: Vegetation impacts and driving conditions. Agricultural and Forest Meteorology, 2017. doi: 10.1016/j.agrformet.2016.08.010

Lotsch, A., Friedl, M.A., Anderson, B.T., Tucker, C.J., Coupled vegetation‐precipitation variability observed from satellite and climate records, Geophysical Research Letters, 30(14), 1774, 2003, doi:10.1029/2003GL017506

Lotsch, A., Friedl, M.A., Anderson, B.T., Tucker, C.J., Response of terrestrial ecosystems to recent Northern Hemispheric drought, Geophysical Research Letters, 32, L06705, 2005, doi:10.1029/2004GL022043

Nemani, R.R., Keeling, C.D., Hashimoto, H., Jolly, W.M., Piper, S.C., Tucker, C.J., Myneni, R.B., Running, S.W.: Climatedriven increases in global terrestrial Net Primary Production from 1982 to 1999, Science, 300(5625), 1560-1563, doi: 515 10.1126/science.1082750, 2003.

Potopová, V., BoroneanÅč, C., Boincean, B., Soukup, J.: Impact of agricultural drought on main crop yields in the Republic of Moldova. Int. J. Climatol., 36, 2063-2082, doi: 10.1002/joc.4481, 2016

S.M. Quiring, S. Ganesh, Evaluating the utility of the Vegetation Condition Index (VCI) for monitoring meteorological drought in Texas, Agric. For. Meteorol., 150, pp. 330-339, 2010

Sidor, C.G., Popa, I., Vlad, R., Cherubini, P.: Different tree-ring responses of Norway spruce to air temperature across an altitudinal gradient in the Eastern Carpathians (Romania), Trees, 29, 985-997, doi: 10.1007/s00468-015-1178-3, 2015.

Toté, C., Swinnen, E., Sterckx, S., Clarijs, D., Quang, C., Maes, R.: Evaluation of the SPOT/VEGETATION Collection 3 reprocessed dataset: Surface reflectances and NDVI, Remote Sensing of Environment, 201, 219-233, 2017, https://doi.org/10.1016/j.rse.2017.09.010.

Trigo R.M., Gouveia C., Barriopedro D., The intense 2007-2009 drought in the Fertile Crescent: Impacts and associated atmospheric circulation. Agricultural and Forest Meteorology, 150, 1245-1257, 2010

Vicente-Serrano, S., Beguería, S., Lorenzo-Lacruz, J., Camarero, J.j., López-Moreno, J.I., Azorin-Molina, C., Revuelto, J., Morán-Tejeda, E., Sanchez-Lorenzo, A.: Performance of drought índices for ecological, agricultural, and hydrological applications. Earth Interact, 16, 1-27, https://doi.org/10.1175/2012EI000434.1, 2012

Vicente-Serrano, S.M., Gouveia, C., Camarero, J.J., Beguería, S., Trigo, R., López-Moreno, J.I,m Azorín-Molina, C., Pasho, E., Lorenzo-Lacruz, J., Revuelto, J., Morán-Tejeda, E., Sanchez-Lorenzo, A.: Response of vegetation to drought timescales across global land biomes. P. Natl. Acad. Sci., 110(1), 52-57, https://doi.org/10.1073/pnas.1207068110, 2013.

Vicente-Serrano, S. M., Camarero, J. J. and Azorin-Molina, C.: Diverse responses of forest growth to drought time-scales in the Northern Hemisphere. Global Ecol. Biogeogr., 23: 1019–1030, doi:10.1111/geb.12183, 2014.

Wang, H., Chen, A., Wang, Q., He, B., Drought dynamics and impacts on vegetation in China from 1982 to 2011, Ecological Engineering, 75, 303-307, 2015. https://doi.org/10.1016/j.ecoleng.2014.11.063
* * *
[Figure]

[Figure]

[Figure]

**Fig. 1.** SPEI time series, averaged over the study region, computed with 3- (top) and 9- (bottom) months' time scale. The red line indicates the -0.84 threshold, used to assess the drought condition

---

## Author Comment (AC3) · 4 Sep 2018

1. The title of the manuscript is a mistake, since the content does not refer to other territory than Romania. Eastern Europe holds a very diverse geography, so that doing a study on Romania and pretending it is about Eastern Europe is huge mistake. Such a market strategy is used to attract public and citations, but it is not a fair approach.

Response: It was not our intention to mislead the readers when choosing this title. The study area encompasses several other countries including the whole Moldova, and parts of Ukraine, Hungary, Serbia, Bulgaria, and Slovakia. However, and as we akcnowledge the reviewer's concern, the title was changed to: "Vegetation vulnerability to drought in Romania and Moldova regions".

[Figure]

2. Lines 185-190: "Negative correlations are more frequently found on the months of May and June" - can you explain why the correlations are more frequent in May and June?

Response: This result is discussed on the Discussion section as follows:

'Levanič et al. (2013), when analysing the relation between Pinus nigra tree-ring widths from southern Romania and monthly precipitation, as well as 3-month SPI, found July to be the month with the highest correlation with precipitation, whereas July, August, and September showed a high correlation with SPI. Moreover, SPEI and NDVI correlations respond to the water balance, and SPEI tends to increase when NDVI decreases (Vicente-Serrano et al., 2013). In the study area, monthly mean precipitation reaches its maximum in June (Koleva et al., 2008; Cheval et al., 2011; Spinoni et al., 2015), and the maximum temperature occurs in July (Spinoni et al., 2015; Dascălu et al., 2016), which points to increased soil water stress that could explain the increase in the correlations between SPEI and NDVI.'

3. The same amendments would be nice for the next paragraphs, as they are mostly descriptive and not enough explanations are provided.

Response: A short explanation will be added in order to clarify the referred issue. We initially decided to discuss the results only in the last section, but taking into account the comments from all the reviewers we will include in the Results section some interpretation, so it won't be as descriptive. Accordingly, we will also improve the Discussion section.

4. Why no correlation in June for "the area south of the Carpathian Mountains" (L194) and "The areas showing positive correlations from April to June are mostly agricultural land" (L197), so including the areas south of the Carpathians?

Response: The sentence 'The areas showing positive correlations from April to June are mostly agricultural land' refers to Figure 3, (as is stated). It is also stated that

this area 'exceeds 37%', which is the value obtained in June at 6 months SPEI. This implies that around 63% of the agricultural land does not present positive correlations in June, and therefore we do not consider that the sentences quoted by the Reviewer are contradictory.

5. Lines 214-219: as mentioned before, the explanations are almost missing, while the very simple description fills the lines. For example, why the standard deviation is lower until July in some forests?

Response: We agree that some details are needed in the results section and accordingly to the comments of the reviewers we will add additional information. In particular, a short explanation will be added in order to clarify the referred issue.

6. The section 3.3 is a very poor description of the drought event 2000/2001, with no explanation and no positioning in the general context of the droughts in Romania, not to say SE Europe as pretended in the title.

Response: The 2000/2001 drought event in SE Europe was chosen for analysis because it is well known and documented, being categorized as one of the worst in the region (http://www.geo.uio.no/edc/droughtdb/edr/DroughtEvents/_2000_Event.php) with strong impacts in several sectors, namely agriculture (Sepulcre-Canto et al., 2012). This drought affected several countries, namely Romania, Hungary and Bulgaria (Sepulcre-Canto et al., 2012), as well as Greece, Turkey and the Balkan countries (http://www.geo.uio.no/edc/droughtdb/edr/DroughtEvents/_2000_Event.php). The extent of this drought episode is depicted on Fig.1, showing the drought classification, as assessed by SPEI 12, across Europe, in February 2001 (http://spei.csic.es/map/maps.html). A short explanation will be added in Introduction and sections 2.4.3 and 3.3 in order to describe with more detail the severe character of the drought event of 2000/2001.

7. The authors claims that "In this section we assess the impact of the strong drought episode of 2000/2001". In reality, the impact has been mentioned very poorly men-

tioned in the section.

Response: In this topic we will clarify in our work that the aim is to analyse and assess the drought (characterized by means of SPEI) impacts on vegetation activity, as obtained using remote sensing data (NDVI). The type of analysis throughout the manuscript was already adopted in several works (Ji and Peters, 2003; Lotsch et al., 2003; 2005; Quiring et al., 2010; Brown et al., 2008; Wang et al., 2015; Vicente Serrano et., 2012; 2013; 2014; Gouveia et al., 2017). The scientific recognition of these methodologies to achieve the goals we proposed lead us to apply this well-known methodology to a very interesting region that we thought lacked studying. Accordingly with our goal and the methodology used in the manuscript, the results described on section 3.3 were obtained considering the months of April to October of the year 2000, corresponding to the months of high vegetation activity. We included two figures (Fig. 9 and 10) that fully support our results. The methodology used allowed to quantify the drought duration and area stricken and presenting vegetation under stress conditions (showing low NDVI anomalies). Fig.9 also allow the evaluation of drought occurrence. Additionally, we also assessed the stress shown on each land cover type in Fig.10. These impacts of drought on vegetation dynamics, namely their duration, frequency and extension, were discussed on the Discussion and Conclusions section on Line 345 Moreover, the analysis of the impacts of drought on different socio-economic sectors, such as agriculture or water storage is not the aim of this work, but will be the purpose of an additional work, now in preparation. However, and acknowledging that the main target of our analysis is misleading, we opted by clarify this issue in the above mentioned sentence, as follow: 'In this section we assess the impact of the strong drought episode of 2000/2001 on vegetation activity, as described using satellite data (NDVI)'

8. This is valid with section 4 too, where the authors claim again that "impacts of droughts on vegetation were analysed". The drought impact is actually indirectly tackled in this paper in the form of the well-known relations between SPEI and NDVI.

Response: Please see answer to previous comment.

References:

Brown, J.F., Wardlow, B.D., Tadesse, T., Hayes, M., J., Reed, B.C., The Vegetation Drought Response Index (VegDRI): A New Integrated Approach for Monitoring Drought Stress in Vegetation, GIScience & Remote Sensing, 45(1), 16-46, 2008

Gouveia, C.M., Trigo, R.M., Beguería, S., Vicente-Serrano, S.M.: Drought impacts on vegetation activity in the Mediterranean region: An assessment using remote sensing data and multi-scale drought indicators, Global Planet. Chan., 151: 15-27. https://doi.org/10.1016/j.gloplacha.2016.06.011, 2017.

Ji, L., Peters, A.J., Assessing vegetation response to drought in the northern Great Plains using vegetation and drought indices, Remote Sensing of Environment, 87(1), 85-98, 2003, https://doi.org/10.1016/S0034-4257(03)00174-3.

Lotsch, A., Friedl, M.A., Anderson, B.T., Tucker, C.J., Coupled vegetation‐precipitation variability observed from satellite and climate records, Geophysical Research Letters, 30(14), 1774, 2003, doi:10.1029/2003GL017506

Lotsch, A., Friedl, M.A., Anderson, B.T., Tucker, C.J., Response of terrestrial ecosystems to recent Northern Hemispheric drought, Geophysical Research Letters, 32, L06705, 2005, doi:10.1029/2004GL022043

Quiring,S.M., Ganesh, S., Evaluating the utility of the Vegetation Condition Index (VCI) for monitoring meteorological drought in Texas, Agric. For. Meteorol., 150, pp. 330-339, 2010

Vicente-Serrano, S., Beguería, S., Lorenzo-Lacruz, J., Camarero, J.j., López-Moreno, J.I., Azorin-Molina, C., Revuelto, J., Morán-Tejeda, E., Sanchez-Lorenzo, A.: Performance of drought índices for ecological, agricultural, and hydrological applications. Earth Interact, 16, 1-27, https://doi.org/10.1175/2012EI000434.1, 2012

Vicente-Serrano, S.M., Gouveia, C., Camarero, J.J., Beguería, S., Trigo, R., López-Moreno, J.I,m Azorín-Molina, C., Pasho, E., Lorenzo-Lacruz, J., Revuelto, J., Morán-Tejeda, E., Sanchez-Lorenzo, A.: Response of vegetation to drought timescales across global land biomes. P. Natl. Acad. Sci., 110(1), 52-57, https://doi.org/10.1073/pnas.1207068110, 2013.

Vicente-Serrano, S. M., Camarero, J. J. and Azorin-Molina, C.: Diverse responses of forest growth to drought time-scales in the Northern Hemisphere. Global Ecol. Biogeogr., 23: 1019–1030, doi:10.1111/geb.12183, 2014.

Wang, H., Chen, A., Wang, Q., He, B., Drought dynamics and impacts on vegetation in China from 1982 to 2011, Ecological Engineering, 75, 303-307, 2015. https://doi.org/10.1016/j.ecoleng.2014.11.063
* * *
[Figure]

[Figure]

**Fig. 1.** SPEI 12 across Europe, in February 2001